# On the Expressive Power of GNNs for Boolean Satisfiability

**Saku Peltonen**
ETH Zürich
Zürich, Switzerland
speltonen@ethz.ch

**Roger Wattenhofer**
ETH Zürich
Zürich, Switzerland
wattenhofer@ethz.ch

## Abstract

Machine learning approaches to solving Boolean Satisfiability (SAT) aim to replace handcrafted heuristics with learning-based models. Graph Neural Networks have emerged as the main architecture for SAT solving, due to the natural graph representation of Boolean formulas. We analyze the expressive power of GNNs for SAT solving through the lens of the Weisfeiler-Leman (WL) test. As our main result, we prove that the full WL hierarchy cannot, in general, distinguish between satisfiable and unsatisfiable instances. We show that indistinguishability under higher-order WL carries over to practical limitations for WL-bounded solvers that set variables sequentially. We further study the expressivity required for several important families of SAT instances, including regular, random and planar instances. To quantify expressivity needs in practice, we conduct experiments on random instances from the G4SAT benchmark and industrial instances from the International SAT Competition. Our results suggest that while random instances are largely distinguishable, industrial instances often require more expressivity to predict a satisfying assignment.

## 1 Introduction

Boolean Satisfiability (SAT) is a central reasoning problem in computer science and one of the canonical NP-complete problems. Classic SAT solvers are highly optimized and can handle instances with millions of variables (Heule et al., 2024). Part of their success comes from carefully engineered heuristics, such as branching and restart strategies (Moskewicz et al., 2001; Biere, 2008), conflict-driven clause learning (CDCL) (Silva & Sakallah, 1996), and learned clause management (Audemard & Simon, 2009). These heuristics are based on recurring patterns that differ across distributions: CDCL solvers are effective on industrial instances with strong community structure (Ansótegui et al., 2012), whereas look-ahead solvers are better suited for random instances (Alyahya et al., 2022). However, designing distribution-specific heuristics is time-consuming and requires expertise in the field.

Machine learning offers a promising alternative, where heuristics can be learned from data. Graph Neural Networks (GNNs) (Scarselli et al., 2008; Kipf, 2016; Xu et al., 2018) have become the main architecture for learning-based SAT solving (Guo et al., 2023), since formulas can be naturally expressed as graphs. A common choice is the Literal Clause Graph (LCG), where literals are connected to clauses in a bipartite graph (see Figure 2 for an example). Existing GNN methods range from end-to-end SAT solvers, such as NeuroSAT (Selsam et al., 2019) and QuerySAT (Ozolins et al., 2022), to hybrid approaches that augment components of classic solvers (Wang et al., 2024; Guo et al., 2023).

However, GNNs are inherently limited in their expressive power (Xu et al., 2019)—that is, their ability to distinguish different graph structures. The expressive power of GNNs is characterized by the Weisfeiler-Leman (WL) test (Weisfeiler & Lehman, 1968), and the extended $k$-WL hierarchy (Immerman & Lander, 1990), which provide universal limits on which graphs GNNs can distinguish. In particular, any GNN bounded by the WL hierarchy produces identical outputs on graphs that are WL-equivalent (Xu et al., 2019). This poses a concrete limitation in the context of SAT,

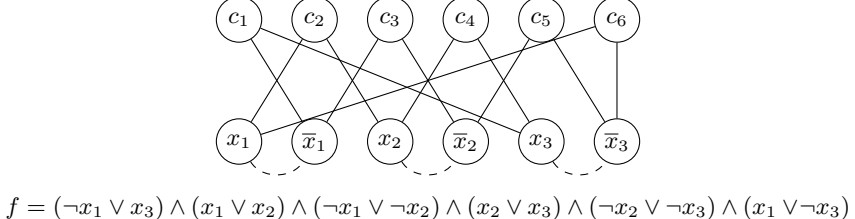

$$f = (\neg x_1 \vee x_3) \wedge (x_1 \vee x_2) \wedge (\neg x_1 \vee \neg x_2) \wedge (x_2 \vee x_3) \wedge (\neg x_2 \vee \neg x_3) \wedge (x_1 \vee \neg x_3)$$

Figure 1: Literal-clause graph with negation connections (LCN) of a formula $f$. Removing the literal-literal edges represented by dashed lines gives the literal-clause graph (LCG).

where solving relies on uncovering structural patterns in graphs representing formulas. This raises a fundamental question: *Are GNNs expressive enough to reason about satisfiability?*

Our main result shows that even the full Weisfeiler-Leman hierarchy cannot, in general, distinguish satisfiable from unsatisfiable formulas. Specifically, we construct pairs of 3-SAT instances with $O(n)$ variables are indistinguishable under the $n$-WL test, despite one being satisfiable and the other unsatisfiable (Theorem 5.3)[1]. This result mirrors the classic construction of Cai et al. (1992) in the context of boolean formulas, demonstrating that indistinguishable formulas may also differ in satisfiability. This has practical implications for solvers that assign variables sequentially, such as QuerySAT (Ozolins et al., 2022). Even with $\Omega(n)$ variable assignments, satisfiability of the residual formula may remain undecidable with a *WL-powerful architecture*[2] (Lemma 5.4, Corollary 5.5). This result is notable because it transfers a theoretical limitation ($k$-WL indistinguishability) to a realistic computational setting.

We analyze regular, planar, and random SAT instances to examine how expressivity requirements vary across families. In RegularSAT—a family which we introduce and show is NP-complete —all instances of the same size are indistinguishable, making WL-powerful GNNs essentially useless. In contrast, PlanarSAT is fully identified by the 4-WL test (Theorem 6.1).

Similarly, we argue that random SAT instances are largely distinguishable by the WL test, mirroring behavior observed in graph isomorphism testing (Babai et al., 1980). We prove this formally for formulas generated using the method of Wu & Ramanujan (2021) (Lemma 6.3). Interestingly, learning-based SAT solvers are often trained on random instances, primarily because they are easy to generate (Li et al., 2024). While both random and industrial instances can be hard in the traditional sense, they differ significantly in structure (Alyahya et al., 2022). Our results show that these instance families also differ in the level of expressivity required to solve them.

We quantify this difference experimentally. We test whether WL-powerful GNNs can theoretically distinguish literals sufficiently to predict a satisfying assignment. For datasets, we use random instances from the G4SAT benchmark (Li et al., 2024) as well as industrial and crafted instances from the International SAT competition (Heule et al., 2024). In general, literals in random instances are quickly distinguished. In contrast, competition instances often require more iterations, and sometimes WL-powerful GNNs are simply not expressive enough to predict a satisfying assignment. This confirms that industrial and crafted instances pose a greater challenge from the perspective of expressivity.

## 2 PRELIMINARIES

**Boolean satisfiability.** Let $x_1, \ldots, x_n$ denote variables in a propositional logic. A literal $\ell$ is a variable $x$ or its negation $\neg x$. A clause $c = \{\ell_1, \ldots, \ell_s\}$ is a set of literals, representing the disjunction $\ell_1 \vee \cdots \vee \ell_s$. A formula $f$ in Conjunctive Normal Form (CNF) is a set of $m$ clauses $\{c_1, \ldots, c_m\}$, representing the conjunction $c_1 \wedge \cdots \wedge c_m$. We write $f$ as a logical formula $\bigwedge_{c \in f} \bigvee_{\ell \in c} \ell$ or as sets, depending on the context. Let $L(f) = \cup_{c \in f} \cup_{\ell \in c} \ell$ be the set of all literals in a formula. The Boolean Satisfiability Problem (SAT) is defined as follows: given a formula $f$, check whether $f$ is

---

[1] A common misconception is that such indistinguishable instances must exist, because SAT is NP-hard. Appendix F explains why expressivity and computational hardness are unrelated in this way.

[2] GNNs with expressive power matching the Weisfeiler-Leman test, such as (Xu et al., 2019)

satisfiable. 3-SAT is the SAT problem where each formula is in CNF and each clause consists of at most 3 literals.

**Graph Notation.** A graph $G$ is a tuple $(V, E)$. If the graph is not clear from context, we write $V(G)$ and $E(G)$. All graphs are undirected unless otherwise specified. $N(v)$ denotes the set of neighbors of a node $v$ and $d(v) = |N(v)|$ is the degree of $v$. On a directed graph $d^{\text{out}}$ denotes the outdegree of a node. A node coloring is a function $\lambda_V : V \to \mathcal{C}$ where $\mathcal{C}$ is a set of colors. Similarly, an edge coloring is a function $\lambda_E : E \to \mathcal{C}$. On edge-colored graphs we write $N_c(v) = \{w \in V(G) : \exists \{v, w\} \in E(G) \text{ s.t. } \lambda_E(\{v, w\}) = c\}$ for the neighbors of $v$ through edges of color $c$.

**Isomorphism.** Two graphs $G$ and $H$ are isomorphic if there exists a bijection $\sigma : V(G) \to V(H)$ such that $\{v, w\} \in E(G)$ iff $\{\sigma(v), \sigma(w)\} \in E(H)$. On graphs with a node-coloring ($c_G, c_H$ for $G, H$, respectively) we also require that $c_G(v) = c_H(\sigma(v))$ for all $v \in V(G)$. The definition is extended for edge-colored graphs in the natural way.

Two CNF formulas $f, g$ are isomorphic if there are bijections $\sigma_L : L(f) \to L(g)$ and $\sigma_C : f \to g$ such that (1.) $\sigma_L(\neg \ell) = \neg \sigma_L(\ell)$ for all $\ell \in L(f)$, i.e. $\sigma_L$ preserves the relationship between a literal and its negation, and (2.) $\sigma_C(c) = \{\sigma_L(\ell) : \ell \in c\}$ for all $c \in f$.

**Graph Neural Networks.** We focus on Message Passing Neural Networks (MPNNs) which encapsulate the majority of GNN architectures. Nodes have some initial features $s_v^0 \in \mathbb{R}^d$. An MPNN operates in synchronous rounds, which are typically structured as follows. In each round $1 \le \ell \le L$, every node aggregates the states of its neighbors, $a_v^\ell = \mathsf{agg}(\{\{s_w^{\ell-1} : w \in N(v)\}\})$, where $\{\{.\}\}$ denotes a multiset. The nodes update their state using their previous state and the aggregated messages: $s_v^\ell = \mathsf{upd}(s_v^{\ell-1}, a_v^\ell)$. The functions $\mathsf{agg}$ and $\mathsf{upd}$ are differentiable functions typically parameterized by neural networks. The final representations $s_v^L$ can be used for node-level prediction tasks, or they can be aggregated into a graph-level representation.

**Weisfeiler-Leman Test.** The expressive power of MPNNs is bounded by the Weisfeiler-Leman (WL) algorithm, also known as color refinement:

**Definition 2.1** (Weisfeiler-Leman algorithm). Let $\lambda_V : V(G) \to \mathcal{C}$ be a vertex coloring. The WL algorithm computes new colorings of the graph iteratively. The initial coloring is given by $\chi^0 := \lambda_V$. For $\ell \in \mathbb{N}$, a new coloring $\chi^\ell$ is defined as $\chi^\ell(v) := (\chi^{\ell-1}(v), \{\{\chi^{\ell-1}(w) : w \in N(v)\}\})$. The color refinement is continued until the partition of nodes given by $\chi^\ell$ equals the partition given by $\chi^{\ell+1}$. The output of the WL algorithm is the stable coloring $\chi^\ell$.

The WL algorithm can be generalized to also use an edge-coloring $\lambda_E : E(G) \to \mathcal{C}_E$. The update considers each color class of edges separately: $\chi^\ell(v) := (\chi^{\ell-1}(v), \{\{\chi_{N_c} : c \in \mathcal{C}_E\}\})$, where $\chi_{N_c} = \{\{\chi^{\ell-1}(w) : w \in N_c(v)\}\}$ are the colors of neighbors through edges of color $c$.

In the *Weisfeiler-Leman test*, the WL algorithm is applied to the disjoint union of $G$ and $G'$. The WL test *distinguishes* $G$ and $G'$ if there is a color $c$ such that the sets $\{v : v \in V(G), \chi(v) = c\}, \{v : v \in V(G'), \chi(v) = c\}$ have different cardinalities. We say that a graph $G$ is *identified* by WL if it is distinguished from every other non-isomorphic graph $H$. A class of graphs $\mathcal{K}$ is identified if every graph in $\mathcal{K}$ is distinguished from every other non-isomorphic graph $H$ (possibly $H \notin \mathcal{K}$).

**k-Weisfeiler Leman Test.** Let $k \ge 2$ be an integer. The *atomic type* of a tuple $\overline{v} \in V(G)^k$ encodes all facts about edge connections and colors within the tuple. Two tuples $\overline{v} \in V(G)^k, \overline{u} \in V(G')^k$ have the same atomic type if and only if the mapping $v_i \mapsto u_i$ is an isomorphism of the induced colored subgraph $G[\{v_1, \ldots, v_k\}]$ to $G[\{u_1, \ldots, u_k\}]$.

**Definition 2.2** ($k$-dimensional WL algorithm). The $k$-Weisfeiler-Leman ($k$-WL) algorithm initializes $\chi^0(\overline{v})$ as the atomic type of $\overline{v}$ for each $\overline{v} \in V(G)^k$. For $\ell \in \mathbb{N}$, a new coloring $\chi^\ell$ is defined as $\chi^\ell(\overline{v}) := (\chi^{\ell-1}(\overline{v}), \chi_1^{\ell-1}(\overline{v}), \chi_2^{\ell-1}(\overline{v}), .., \chi_k^{\ell-1}(\overline{v}))$, where $\chi_i^{\ell-1}(\overline{v}) = \{\{\chi^{\ell-1}(\overline{v}_1, \ldots, \overline{v}_{i-1}, u, \overline{v}_{i+1}, \ldots, \overline{v}_k) : u \in V(G)\}\}$. The color refinement is continued until the partition of tuples given by $\chi^\ell$ equals the partition given by $\chi^{\ell+1}$. The output of the WL algorithm is the stable coloring $\chi^\ell$. The $k$-dimensional WL test is defined analogously to the WL test. We say that $G$ and $G'$ are distinguished if there exists a color $c$ such that the sets $\{\overline{v} : \overline{v} \in V(G)^k, \chi(\overline{v}) = c\}$ and $\{\overline{v} : \overline{v} \in V(G'), \chi(\overline{v}) = c\}$ have different cardinalities.

*Remark* 2.3. There are two algorithms and naming conventions in the literature. This version of the $k$-WL algorithm is most common in machine learning literature. Through connections to counting

logic, it can be shown (Grohe, 2017) to be equivalent to $(k-1)$-WL, as defined in for example (Cai et al., 1992; Kiefer et al., 2019). See (Huang & Villar, 2021) for an overview.

## 3 GRAPH REPRESENTATIONS OF SAT FORMULAS

A standard way to represent SAT formulas as graphs is the *literal-clause graph* (LCG). The LCG is a bipartite graph with literals on one side and clauses on the other. Edges connect literals to clauses where they appear. See Figure 1 for an example.

In GNNs, node labels of a graph are omitted to preserve permutation invariance. To avoid information loss, it is essential to include edges between literals and their negations.[3] We call the extended representation the *literal-clause graph with negation connections* (LCN). It is defined as the LCG with additional edges connecting each variable to its negation. The literal-literal edges are assigned a distinct color from the literal-clause edges. The LCN of a CNF formula $f$ is denoted $\mathrm{LCN}(f)$.

Literal-literal edges is necessary to preserve information once labels are removed (see Appendix A for an example). The LCN representation is also sufficient to preserve all information:

*Observation* 3.1. An LCN without node labels uniquely determines the corresponding SAT formula up to isomorphism. The formula can be constructed by grouping nodes into variable pairs according to the literal-literal edges, labeling them arbitrarily as $x_i, \neg x_i$, and reading clauses from the literal-clause edges.

Other representations include the *variable-clause graph* (VCG) and less common *literal-incidence graph* (LIG) and *clause-incidence graph* (CIG). We focus on the LCN in this work, as the LIG and CIG representations are lossy, and VCG is not suitable for expressivity analysis because the same formula (up to isomorphism) can have non-isomorphic VCG representations. See Appendix A for details.

## 4 RELATED WORK

**Machine Learning for SAT.** One of the earliest works in this area is NeuroSAT (Selsam et al., 2019) – an end-to-end SAT solver framework with GNNs. Their algorithm is based on predicting satisfiability, and hence works with single-bit supervision. QuerySAT (Ozolins et al., 2022) uses an unsupervised loss computed from continuous variable values, and a query mechanism to update the variable values. Other approaches include transformer-based models such as SATformer (Shi et al., 2023) and attention-based variants like SAT-GATv2 (Chang & Liu, 2025). In a complementary line of work, Yolcu & Póczos (2019) build a local search SAT solver whose variable selection strategy is learned by a GNN. Several of these architectures and losses can be evaluated on the G4SAT benchmark (Li et al., 2024).

The end-to-end SAT solvers are mostly of methodological interest, and currently not practical for large instances. Another line of research augments classic SAT solvers with machine learning components, such as learned heuristics or branching strategies. Selsam & Bjørner (2019) adapt the NeuroSAT architecture to predict unsatisfiability cores, which is used to select branching variables. Other SAT solving components with potential for ML solutions include variable initialization (Wu, 2017), clause deletion (Vaezipoor et al., 2020) and restart policy (Liang et al., 2018b). See (Guo et al., 2023) for a comprehensive survey on machine learning methods in SAT solving.

Dataset generation is another promising application area. To mimic industrial SAT instances, Wu & Ramanujan (2021) use a learning-based graph representation and design a method to generate SAT instances from their implicit model. Another line of work frames SAT generation as a bipartite graph generation problem (You et al., 2019).

**Expressivity and the Weisfeiler-Leman test.** It is well known that the Weisfeiler-Leman test bounds the expressive power of MPNNs (Xu et al., 2019). This limitation has motivated a large number of more expressive GNN architectures, with expressivity corresponding to $k$-WL for some

---

[3]Literal-literal edges are already used in practice in most GNN SAT solvers (Selsam et al., 2019; Li et al., 2024), but their importance is not always stated explicitly.

$k > 2$ (Morris et al., 2019; Maron et al., 2019; Keriven & Peyré, 2019). A comparison of the expressivity of different GNN extensions is given by (Papp & Wattenhofer, 2022).

The $k$-WL is a powerful tool, but it is not able to solve graph isomorphism in general. The seminal work of Cai et al. (1992) shows that there are pairs of non-isomorphic $O(n)$-node graphs that are indistinguishable by the $n$-WL test. There are positive results for special graph classes. Namely, Kiefer et al. (2019) show that planar graphs are identified by 4-WL. Random graphs are mostly identifiable by the WL test in two iterations (Babai et al., 1980).

Beyond structural expressivity, Grohe (2023) analyzes the power of GNNs in terms of circuit complexity, showing that GNNs can decide problems in $\text{TC}^0$ (constant-depth circuits with polynomial size).[4]

**Complexity Theory.** Boolean satisfiability was the first problem proven to be NP-complete, by Stephen Cook (Cook, 1971). Later, several variants of SAT have been proven to be equally hard, such as PlanarSAT (Lichtenstein, 1982). SAT solving remains an active area of research, with SAT competitions being held annually (Heule et al., 2024).

The computational complexity of different equivalence relations between boolean functions was studied by Borchert et al. (1998). In their work, two formulas $f, f'$ on the same set of variables are said to be isomorphic if there is a bijection $\sigma$ of the variables such that $f, f'$ agree on all assignments up to the mapping $\sigma$. Under this definition, for instance, a tautology is isomorphic to the empty formula. Our notion of isomorphism is stricter, as it requires clauses to be preserved under the mapping. We find that this notion is better suited for the setting with graph representations.

**Proof Complexity.** The complexity of proving the unsatisfiability of propositional formulas is a central topic in proof complexity. One of the most studied systems is resolution, where the proof consists of clauses derived from the original formula using a simple inference rule. Hard examples for resolution include the Pigeonhole principle (Haken, 1985) and the Tseitin formulas (Urquhart, 1987). Resolution proof length can be related to the width of the proof, where the width of a resolution proof is the maximum number of literals in any clause of the proof (Ben-Sasson & Wigderson, 2001). The complexity of resolution has also been characterized in terms of pebbling games (Atserias & Dalmau, 2008; Galesi & Thapen, 2005). In this setting, pebbling games are played on a single graph—unlike the two-graph pebbling games that correspond to $k$-WL indistinguishability (Cai et al., 1992).

## 5 INDISTINGUISHABLE FAMILIES OF SAT INSTANCES

In this section, we construct explicit families of SAT formulas that are provably indistinguishable by the WL-test and the WL hierarchy. As our main technical contribution, we show that there are 3-SAT formulas that are indistinguishable by the $n$-WL test, despite one being satisfiable and the other not (Section 5.2). We also identify a practically relevant family of *regular* SAT formulas that are indistinguishable by WL, yet remain NP-complete. In general, distinguishing SAT instances (regardless of satisfiability) is as hard as graph isomorphism (Appendix C).

### 5.1 3-REGULAR SAT

To motivate our first contribution, we start the section with a simple example of a pair of WL-indistinguishable formulas. Consider a CNF formula $f$ on three variables $x_1, x_2, x_3$:

$$\begin{aligned} f = (x_1 \vee \neg x_3) \wedge (\neg x_1 \vee x_3) \ldots && x_1 = x_3 \\ \wedge (x_1 \vee x_2) \wedge (\neg x_1 \vee \neg x_2) \ldots && x_1 \otimes x_2 \\ \wedge (x_2 \vee x_3) \wedge (\neg x_2 \vee \neg x_3) && x_2 \otimes x_3 \end{aligned}$$

where $\otimes$ denotes the xor. See Figure 1 for the graph representation. The formula is satisfied by $x = (1, 0, 1)$ or $x = (0, 1, 0)$. We can make a similar but unsatisfiable formula $f'$ by replacing the

---

[4]This implies (under common complexity theoretic assumptions) that SAT cannot be decided by GNNs. However, note that this does not imply that there must exist $n$-WL indistinguishable satisfiable and unsatisfiable formulas (which is what we show in Theorem 5.3). Indeed, as shown in Theorem 6.1, all PlanarSAT instances are distinguishable by 4-WL, even though PlanarSAT is NP-complete.

clauses encoding $x_2 \otimes x_3$ (the third line) with clauses encoding $x_2 = x_3$. Note that this change keeps all literal degrees the same. Since each literal appears in exactly two clauses in both $f$ and $f'$, the LCGs of $f$ and $f'$ are WL-indistinguishable. However, $f'$ is clearly unsatisfiable.

This example can be generalized to a family of 3-regular SAT instances. We say that a SAT instance is *k-regular* if each literal appears in exactly $k$ clauses and each clause contains exactly $k$ literals. Despite this strong regularity, the class remains computationally hard:

**Theorem 5.1.** *3-regular* SAT *is* NP-complete.

Although related NP-complete variants have appeared in the literature (3-SAT (Karp, 1972), 3-SAT with each *variable* appearing at most 4 times (Tovey, 1984)), we are not aware of a formal proof of this specific result, so we provide one in Appendix D for completeness. If formulas are given in the 3-regular SAT format, a WL-powerful GNN is essentially useless in solving them:

*Observation* 5.2. The WL test does not distinguish between any two 3-regular SAT formulas with the same number of variables.

## 5.2 K-WL INDISTINGUISHABLE SAT INSTANCES

Given that WL cannot distinguish some formulas, one might wonder whether higher-order WL tests suffice. In this section, we answer the question in the negative, showing that there are 3-SAT formulas that are indistinguishable by the WL hierarchy, despite one being satisfiable and the other not.

**Theorem 5.3.** *There are* 3-SAT *formulas* $f, \tilde{f}$ *with* $O(n)$ *variables and* $O(n)$ *clauses such that* $f$ *is satisfiable and* $\tilde{f}$ *is not, but the* LCN*s of* $f$ *and* $\tilde{f}$ *are indistinguishable by the* $n$-WL *test.*

Our result uses the seminal work of Cai, Fürer and Immerman Cai et al. (1992), giving a pair of non-isomorphic graphs $H$ and $\tilde{H}$ which are indistinguishable by $n$-WL. We construct a pair of formulas $f, \tilde{f}$ with LCNs isomorphic to $H$ and $\tilde{H}$, respectively. On a high level, our formula $f_G$ encodes the existence of an *even orientation* for a graph $G$. This is an orientation of the edges such that each node has an even number of outedges. We show that such an orientation exists if and only if the number of edges is even. Then, we construct a *twisted* formula $\tilde{f}_G$, encoding the existence of an even orientation when one of the edges is bidirectional. Exactly one of $f_G$ and $\tilde{f}_G$ are satisfiable, depending on the parity of $m$. The proof of Theorem 5.3 is given in Appendix B.

Interestingly, the construction in Theorem 5.3 is similar to Tseitin formulas, which are known as hard instances for resolution refutation (Urquhart, 1987). Tseitin formulas encode a set of linear inequalities over nodes of a graph. See Definition B.11 for a formal definition. Resolution proofs—and likewise the WL test—rely on exploiting local patterns in the formula (or graph), and in both settings, the hardest instances include a global inconsistency that cannot be detected through purely local reasoning. To the best of our knowledge, this connection between Tseitin formulas and the construction of Cai et al. (1992) has not been previously observed.

**Implications for WL-powerful architectures.** Theorem 5.3 shows that in general, satisfiability cannot be expressed by any GNN architecture bounded by the WL hierarchy. In contrast to predicting satisfiability directly, classic SAT solvers, and some GNN-based approaches (Ozolins et al., 2022), work by sequentially setting variables. A natural question is whether a few variable assignments can help to break symmetries, making formulas more distinguishable, i.e., increasing the effective expressive power of the solver. This mirrors how node *labeling tricks* are useful for GNNs to solve certain tasks like triangle counting (You et al., 2021; Zhang et al., 2021). However, even in this setting, WL-powerful GNNs have fundamental limitations:

**Lemma 5.4.** *Let* $f, \tilde{f}$ *be formulas with* LCN*s indistinguishable by* $k$-WL *for some* $k \geq 4$. *For any partial assignment* $\sigma$ *of variables of* $f$ *with at most* $\lfloor k/2 \rfloor - 1$ *variables set, there is a corresponding partial assignment* $\tilde{\sigma}$ *of the variables of* $\tilde{f}$ *such that* $\mathrm{LCN}(\sigma(f))$ *and* $\mathrm{LCN}(\tilde{\sigma}(\tilde{f}))$ *are* WL-*indistinguishable.*

Here, $\mathrm{LCN}(\sigma(f))$ denotes the LCN of the formula $f$, with additional labels $\top$ or $\bot$ for literals set to true or false by $\sigma$. Lemma 5.4 transfers a theoretical $k$-WL indistinguishability result to a practical setting, where a WL-powerful GNN is used to make sequential decisions—any decision that is made for $f$ could also be made for $\tilde{f}$.

Restart strategies provide a concrete example of this limitation. When restarting, the solver discards the current assignment and restarts the search from scratch. Restarting is a core component of classic solvers (Gomes et al., 2000; Huang et al., 2007), and ML has also been used to guide such restarts (Liang et al., 2018a). The following result is a corollary of Theorem 5.3 and Lemma 5.4, showing that restart prediction is fundamentally hard for WL-powerful models:

**Corollary 5.5.** *WL-powerful GNNs cannot, in general, distinguish a satisfiable residual formula from an unsatisfiable one, even with $\Theta(n)$ variable assignments, where $n$ is the number of variables in the formula.*

## 6 POSITIVE RESULTS FOR DISTINGUISHABILITY

Having shown expressivity limits, we now identify families where GNNs can succeed.

### 6.1 PLANAR SAT

PlanarSAT is a variant of SAT where the clauses are represented as a planar graph. The PlanarSAT language is NP-complete (Lichtenstein, 1982). The following result is a consequence of Kiefer et al. (2019), who showed that the 4-WL test distinguishes all planar graphs.

**Theorem 6.1.** *For any* SAT *formula $f$, there is an equisatisfiable* PlanarSAT *formula $f'$ with polynomially many variables and clauses, such that the 4-WL test distinguishes $f'$ from any other formula.*

*Proof.* PlanarSAT is NP-complete (Lichtenstein, 1982) (this also works in the LCN representation, see Lemma 1 in (Lichtenstein, 1982)). The 4-WL test distinguishes between all planar graphs (Kiefer et al., 2019). □

This result shows that, despite the general limitations of the WL-hierarchy in distinguishing formulas (Section 5.2), there exist natural and computationally hard subsets of SAT, such as PlanarSAT, where already 4-WL is fully expressive. The reduction to PlanarSAT is done by replacing edge crossings in the LCN with gadgets that ensure planarity. Unfortunately, the reduction is not efficient in practice, because each gadget adds 9 variables and 20 clauses and there may be up to $O(n^2)$ edge crossings.

### 6.2 RANDOM SAT INSTANCES

Random SAT instances can be defined in various ways, often depending on the desired structure or difficulty. In this section, we consider instances generated from randomly sampled literal-incidence graphs (LIGs), where each literal is a node and edges connect literals that co-occur in a clause. While the LIG representation loses some logical information, Wu & Ramanujan (2021) give a principled procedure for extracting a CNF formula from it:

**Lemma 6.2** ((Wu & Ramanujan, 2021)). *Given a literal-incidence graph $G$, a corresponding* CNF *formula can be extracted by computing a minimal clique edge cover of $G$.[5] The clauses of the formula correspond to the cliques in the edge cover. The generated formula does not contain duplicate clauses, subsumed clauses or unit clauses.*

We show that a CNF formula extracted from a random literal-incidence graph is likely identified by WL. The proof can be found in Appendix E.

**Theorem 6.3.** *A CNF formula extracted from a uniformly random literal-incidence graph with $n$ literals is identified by the WL test with probability at least $1 - (n)^{-1/7}$, over the choice of a LIG, for a large-enough $n$.*

## 7 EXPERIMENTS

We aim to evaluate whether WL-powerful architectures (such as GIN (Xu et al., 2019)) are, in principle, capable of predicting a satisfying assignment to SAT formulas.

---

[5]A clique edge cover is a set of cliques in $G$, such that all edges belong to at least one clique.

## 7.1 SETUP

Our experiments are based on the fact that in a node-level prediction task, nodes that are WL-equivalent must have the same output. Given a satisfiable formula, we add equality constraints between all WL-equivalent literals, and check whether the augmented formula remains satisfiable. This is a necessary (but not a sufficient) condition for a WL-powerful GNN to predict a satisfying assignment.

Formally, let $f$ be a satisfiable formula. Running WL for $r \geq 1$ rounds on $\mathrm{LCN}(f)$ gives a partition of the literals $L_1, \ldots, L_s$. We construct an augmented formula $f_r$ that restricts literals in each partition to the same value. For each equivalence class $L_j = \{\ell_1^j, \ldots, \ell_{n_j}^j\}$, we add the clauses $g_j := (\neg \ell_{n_j}^j \vee \ell_1^j) \wedge \bigwedge_{i=1}^{n_j-1} (\neg \ell_i^j \vee \ell_{i+1}^j)$. The formula $g_j$ encodes an equality constraint between the literals in $L_j$. Given a satisfiable formula $f$, a WL-powerful architecture can predict a satisfying assignment within $r$ rounds *only if* $f_r = f \wedge \bigwedge_{i=1}^{s} g_j$ is satisfiable. We solve $f_r$ for different values of $r$, from $r = 1$ to $r = r_{\mathrm{converged}}$, where $r_{\mathrm{converged}}$ is the number of rounds for WL to converge on the LCN of the original formula $f$. We let $r_{\mathrm{crit}}$ be the smallest $r$ such that $f_r$ is satisfiable, if such a round exists. Conversely, if $f_r$ is unsatisfiable for all $r$, we conclude that WL is not sufficiently powerful to predict satisfying assignments for $f$.

## 7.2 DATASETS

**Random instances.** The G4SAT benchmark (Li et al., 2024) generates random instances from various families, including random 3-SAT (Crawford & Auton, 1996) and the *CA* family mimicking community structures in industrial instances (Giráldez-Cru & Levy, 2015). Instances are grouped by size (referred to as difficulty in (Li et al., 2024)). The benchmark is designed so that all instances are challenging, by choosing instances that are near the satisfiability threshold for 3-SAT (Crawford & Auton, 1996) and analogously for other families. See (Li et al., 2024) for descriptions of the families.

**SAT competition instances.** We use instances from the International SAT competition (Heule et al., 2024), spanning the competitions held between years 2020 and 2025. The instances are selected as hard examples from various applications, including scheduling, cryptography, and hardware equivalence checking. Some families are hand-crafted to be difficult for SAT solvers, such as formulas encoding the Pigeonhole principle. Detailed descriptions of the families can be found in the competition proceedings; see (Heule et al., 2024) for the 2024 edition.[6] The size of the instance varies from a few hundred variables to 50 million variables. Due to limited computational resources, we limit instances to those under 10 MB in size.

## 7.3 RESULTS

See Table 1 for results on random instances. The number of rounds needed for WL to distinguish literals sufficiently is very low, usually 3 or 4. We observed that in many cases (about 40% of all formulas), WL actually gives all literals unique identifiers. In this case, the constrained formula $f_r$ is trivially satisfiable because it is equal to $f$. There are a few instances in the $k$-clique and $k$-vercov families (2.0% and 0.3%, respectively) that WL could not solve.[7] See Table 4 for the full results on random instances.

For random 3-SAT instances, regardless of the size of the formula, literals are almost always sufficiently identified after 3 iterations, and WL converges in 4 rounds. This pattern is due to the constant degree of the clauses. In the first iteration, each literal sees its degree $d_\ell$. However, the second iteration does not refine the literal partition because every neighbor is a clause with degree 3—only on the third iteration, the literals observe the degrees of other literals in shared clauses.

---

[6] A list of instances for the 2024 competition is available at `https://benchmark-database.de/?track=main_2024&result=sat`

[7] $k$-clique encodes the problem of finding a $k$-clique on an Erdös-Renyi graph. Symmetries in the underlying graph can lead to symmetries in the formula that WL cannot resolve. For example, the graph may contain two nodes that are each fully connected to the same clique but not to each other, making them indistinguishable under WL, yet mutually exclusive in the satisfying assignment.

Table 1: Results on random instances, grouped by family. All instances were initially satisfiable. $r_{\text{crit}}$ is the smallest number of rounds for which the augmented formula $f_r$ is satisfiable. We write *unsat* when this value does not exist. $r_{\text{converged}}$ is the number of rounds for WL to converge. All values are reported as mean ± std.

| family | difficulty | $r_{\text{crit}}$ | $r_{\text{converged}}$ | Variables | Clauses | Count |
|---|---|---|---|---|---|---|
| 3-sat | easy | 2.97 ± 0.18 | 3.68 ± 0.47 | 26 ± 9 | 119 ± 36 | 1000 |
| | medium | 3.00 ± 0.04 | 3.92 ± 0.28 | 119 ± 47 | 509 ± 198 | 1000 |
| | hard | 3.00 ± 0.00 | 4.00 ± 0.00 | 250 ± 29 | 1065 ± 125 | 100 |
| | hard+ | 3.00 ± 0.00 | 4.00 ± 0.00 | 921 ± 48 | 3775 ± 196 | 24 |
| | hard++ | 3.08 ± 0.28 | 4.00 ± 0.00 | 5001 ± 62 | 20504 ± 256 | 25 |
| k-clique | easy | 4.12 ± 0.73 | 6.26 ± 0.83 | 33 ± 13 | 543 ± 426 | 960 |
| | | unsat | 6.00 ± 0.78 | 22 ± 7 | 217 ± 194 | 40 |
| | medium | 4.11 ± 0.52 | 6.33 ± 0.95 | 68 ± 17 | 2156 ± 960 | 999 |
| | | unsat | 6.00 ± 0.00 | 45 ± 0 | 939 ± 0 | 1 |
| | hard | 4.00 ± 0.00 | 6.00 ± 0.00 | 114 ± 20 | 5554 ± 1718 | 100 |

Table 2: Results on a selection of instances from the international SAT competition. All instances were initially satisfiable.

| family | $r_{\text{crit}}$ | $r_{\text{converged}}$ | Variables | Count |
|---|---|---|---|---|
| argumentation | 2.94 ± 0.44 | 4.31 ± 0.87 | 1266 ± 625 | 16 |
| circuit-multiplier | unsat | 7.18 ± 0.40 | 1075 ± 50 | 11 |
| cryptography | 15.74 ± 14.67 | 17.63 ± 14.34 | 41510 ± 29705 | 19 |
| | unsat | 18.74 ± 13.38 | 19257 ± 37634 | 23 |
| hamiltonian | 4.17 ± 0.51 | 5.44 ± 0.51 | 550 ± 45 | 18 |
| heule-folkman | unsat | 4.91 ± 0.30 | 16614 ± 1103 | 11 |
| heule-nol | unsat | 8.60 ± 1.26 | 1419 ± 0 | 10 |
| maxsat-optimum | 26.64 ± 3.64 | 29.27 ± 2.49 | 22157 ± 5623 | 11 |
| | unsat | 59.00 ± 0.00 | 27597 ± 8713 | 7 |

**SAT Competition Instances.** An overview of the results is shown in Table 2 and a more detailed breakdown is given in Table 3 (Appendix G). WL takes considerably more rounds to converge. Out of 448 evaluated instances, only 234 could be solved with the expressive power of WL. Across 69 instance families, 38 contained instances where WL is not expressive enough. An example of a family with indistinguishable structure is *heule-nol*, which encodes a type of grid coloring problem (Heule et al., 2024). The regular structure of the instances makes it difficult for WL to distinguish literals.

## 8 CONCLUSIONS AND FUTURE WORK

Our theoretical results establish fundamental limitations on the expressive power of GNNs for SAT solving. We show that even the full WL hierarchy cannot distinguish between satisfiable and unsatisfiable formulas, while also revealing connections to resolution complexity and offering a new perspective on the classic construction of Cai et al. (1992). Additionally, we identify an NP-complete but WL-indistinguishable class of SAT instances, as well as provide positive guarantees for distinguishing random and planar SAT instances.

Experimentally, we show that WL-powerful architectures are, in principle, expressive enough to predict satisfying assignments for random SAT instances, but struggle with industrial and crafted benchmarks. Our test setup allows us to see if a WL-powerful GNN has the *necessary* expressive power for predicting satisfying assignments, though it does not capture whether that expressivity is *sufficient* for generalizable learning. Even for random SAT formulas, improved generalization may require higher-order GNNs, symmetry breaking, or other architectural improvements.

We hope to see GNNs applied to industrial SAT instances in the future. While this remains challenging—due to the lack of scalable generators and the large size of many industrial instances—these instances provide a structurally richer and potentially more demanding testbed. Progress in this direction could offer new insights into generalization that remain hidden when only using random instance distributions.

## REPRODUCIBILITY STATEMENT

The code for the experiments is available online[8] Complete proofs of all theoretical results, including background results such as the NP-completeness of 3-regular SAT, are included in the appendix (Appendices B-E). Details of dataset selection and generation are given in Section 7.2 and Appendix G.

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

## A GRAPH REPRESENTATIONS OF SAT FORMULAS

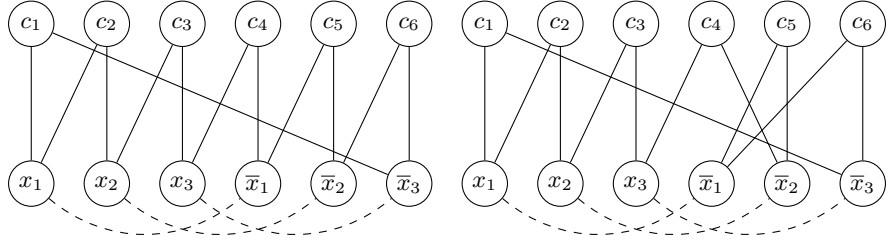

Left: $f = (x_1 \lor \neg x_3) \land (x_1 \lor x_2) \land (x_2 \lor x_3) \land (x_3 \lor \neg x_1) \land (\neg x_1 \lor \neg x_2) \land (\neg x_2 \lor \neg x_3)$
Right: $f' = (x_1 \lor \neg x_3) \land (x_1 \lor x_2) \land (x_2 \lor x_3) \land (x_3 \lor \neg \mathbf{x_2}) \land (\neg x_2 \lor \neg x_1) \land (\neg \mathbf{x_1} \lor \neg x_3)$

Figure 2: LCNs of $f$ and $f'$. The difference between the formulas is highlighted in bold. Removing the literal-literal edges represented by the dashed lines gives the literal-clause graphs.

**Literal-clause graphs with negation connections** See Section 3 for the definition of the literal-clause graph with negation connections (LCN). The following lemma illustrates why adding literal-literal edges is necessary to preserve all information about the formula when labels are removed.

**Lemma A.1.** *There are* 3-SAT *formulas* $f, f'$ *such that the literal-clause graphs* $G_f$ *and* $G_{f'}$ *are isomorphic but* $f$ *is satisfiable and* $f'$ *is not.*

*Proof.* Consider the formulas shown in Figure 2 The two LCGs (dashed lines excluded) are isomorphic. $f$ has a solution $x_1 = 1, x_2 = 0, x_3 = 1$. However, $f'$ is not satisfiable: $x_1 = 1$ implies $x_2 = 0$ because of $c_5$, and $x_3 = 0$ because of $c_6$. Now $c_3$ is false. Conversely, $x_1 = 0$ implies $x_2 = 1$ because of $c_2$, and $x_3 = 0$ because of $c_1$. This makes $c_4$ false. $\square$

**Other representations** In this work we use the literal-clause graph representation (with negation edges) because it is lossless and corresponds one-to-one with formulas up to isomorphism. Another popular representation is the *variable-clause graph* (VCG), which connects variables to clauses, with edge colors indicating the sign of the variable. Similarly to the LCN, the VCG preserves all information about the formula. However, a limitation of the VCG representation is that a formula admits up to $2^n$ non-isomorphic graph encodings, where $n$ is the number of variables. For example, flipping the sign of all occurrences of a variable $x$ in $f = (x \vee y) \wedge (\neg x)$ produces an isomorphic formula $f' = (\neg x \vee \neg y) \wedge (x)$, but the VCGs are non-isomorphic. In contrast, the LCNs of $f$ and $f'$ are identical. From the perspective of learning-based methods, such invariance is desirable: it is analogous to the requirement of permutation invariance in graph learning.

Other known graph representations include the *literal-incidence graph* (LIG) and the *clause-incidence graph* (CIG), where literals (clauses) are connected to other literals (clauses) if they co-occur in a clause (share a variable). These representations are less commonly used, as they are inherently lossy.

## B   K-WL INDISTINGUISHABLE INSTANCES (PROOF OF THEOREM 5.3)

In this section, we prove the following theorem:

**Theorem 5.3.** *There are* 3-SAT *formulas* $f, \tilde{f}$ *with* $O(n)$ *variables and* $O(n)$ *clauses such that* $f$ *is satisfiable and* $\tilde{f}$ *is not, but the* LCN*s of* $f$ *and* $\tilde{f}$ *are indistinguishable by the* $n$-WL *test.*

The construction is based on the seminal work of Cai, Fürer, Immerman (CFI) (Cai et al., 1992), giving a pair of non-isomorphic graphs $H$ and $\tilde{H}$ which are indistinguishable by $n$-WL. We construct a pair of formulas $f, \tilde{f}$ with LCNs isomorphic to $H$ and $\tilde{H}$, respectively. On a high level, our formula $f_G$ encodes the existence of an *even orientation* for a graph $G$. This is an orientation of the edges such that each node has an even number of outedges. We show that such an orientation exists if and only if the number of edges is even. Then, we construct a *twisted* formula $\tilde{f}_G$, encoding the existence of an even orientation when one of the edges is bidirectional. Exactly one of $f_G$ and $\tilde{f}_G$ are satisfiable, depending on the parity of $m$.

### B.1   THE CONSTRUCTION

The CFI construction (Cai et al., 1992) takes in a low degree graph $G$ with only linear sized separators and produces non-isomorphic graphs $X(G)$ and $\tilde{X}(G)$ that are indistinguishable by WL. Next, we go over the steps to construct $X(G)$, or equivalently a formula $f_G$ with an LCN isomorphic to $X(G)$. For each vertex $v \in V(G)$, we define the following subformula $X_k$, where $k = d(v)$:

$$
\begin{aligned}
&\text{Literals: } A_k \cup B_k, \text{ where} \\
&A_k = \{a_i \mid 1 \le i \le k\}, \\
&B_k = \{b_i \mid 1 \le i \le k\} \\
&\text{Clauses: } C_k \cup D_k, \text{ where} \\
&C_k = \{c_S = \left(\vee_{i \in S}\, a_i\right) \vee \left(\vee_{i \notin S}\, b_i\right) \mid S \subseteq [k], |S| \text{ is even}\} \\
&D_k = \{a_i \vee b_i \mid i \in \{1, \ldots, k\}\}
\end{aligned}
$$

Here $[k]$ denotes the set $\{1, \ldots, k\}$. See Figure 3 for a diagram of the LCN of $X_k$ with $k = 3$. This graph corresponds exactly to the graph $X_k$ in the CFI construction[9]. Note that the negations of the literals are not contained in $X_k$ – they will be defined later.

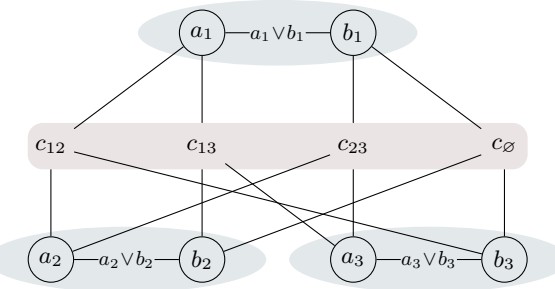

Figure 3: An LCN of the formula $X_3$, corresponding to the graph $X_3$ in the CFI construction. Literal nodes are circled. The clauses are connected to the literals by solid lines.

For a given graph $G$, the full formula $f_G$ is constructed as follows. For each vertex $v \in V(G)$, add the subformula $X_{d(v)}$. Each edge $\{v, w\}$ of $v$ is associated with one of the literal pairs $(a_i, b_i)$, where we call the literals $a_{v,w}, b_{v,w}$. The node $w$ on the other side of this edge uses the negations of the literals, that is, $a_{w,v} = \neg a_{v,w}$ and $b_{w,v} = \neg b_{v,w}$. In the LCN, the literal $a_{v,w}$ is connected to its negation $a_{w,v}$, and $b_{v,w}$ to $b_{w,v}$. See Figure 4a for an example.

Now, we define a *twisted* formula $\tilde{f}_G$, and the corresponding twisted graph $\tilde{X}(G)$ as follows. An edge $\{v, w\} \in E(G)$ is chosen arbitrarily. The literal-literal connections are twisted, so that $a_{v,w}$ becomes the negation of $b_{w,v}$ and $b_{v,w}$ becomes the negation of $a_{w,v}$. A twisted edge is shown in Figure 4b.

*Observation* B.1. The LCNs of $f_G$ and $\tilde{f}_G$ are isomorphic to the graphs $X(G)$ and $\tilde{X}(G)$ in (Cai et al., 1992), respectively.

To state the result of Cai et al. (1992), we need the concept of a separator:

**Definition B.2.** A *separator* of a graph $G$ is a set $S \subset V(G)$ such that the induced subgraph on $V \setminus S$ has no connected component with more than $|V(G)|/2$ vertices.

**Theorem B.3** (Theorem 6.4 in (Cai et al., 1992)). *Let $G$ be a graph such that every separator of $G$ has at least $k + 1$ vertices. Then $X(G)$ and $\tilde{X}(G)$ are non-isomorphic but k-WL indistinguishable.*

### B.2 SATISFIABILITY OF $f_G$ AND $\tilde{f}_G$

By construction, literals have their negations on the other side of each edge:
*Remark* B.4. For a normal (not twisted) edge $\{v, w\} \in E(G)$, $a_{v,w} = \neg a_{w,v}$ and $b_{v,w} = \neg b_{w,v}$. If the edge is twisted, $a_{v,w} = \neg b_{w,v}$ and $b_{v,w} = \neg a_{w,v}$.

The following is true for $f_G$ and $\tilde{f}_G$, for any edge $\{v, w\}$ (twisted or not):
*Observation* B.5. The literals $a_{v,w}, b_{v,w}$ are non-equal in any satisfying assignment.

*Proof.* This is forced by $(a_{v,w} \vee b_{v,w}) \wedge (a_{w,v} \vee b_{w,v}) \equiv (a_{v,w} \vee b_{v,w}) \wedge (\neg a_{v,w} \vee \neg b_{v,w})$. $\square$

This allows us to talk about satisfying assignments of $f_G$ and $\tilde{f}_G$ as orientations of edges of $G$:
*Remark* B.6. Any satisfying assignment is uniquely characterized by the values of $A$. On a graph without twists, assignments to $A$ correspond one-to-one with orientations of $G$. On a twisted edge, either both $a_{v,w}$ and $a_{w,v}$ are true, or both are false.

Consequently, the number of $A$-literals set to true in $f_G$ is $m$, while in $\tilde{f}_G$ it is $m - 1$ or $m + 1$.

The next lemma characterizes the satisfiability of $X_k$:

---

[9]The nodes corresponding to the $D_k$ clauses are not present in the standard construction in (Cai et al., 1992), but they mention that nodes connecting each $a_i$ to $b_i$ can be added.

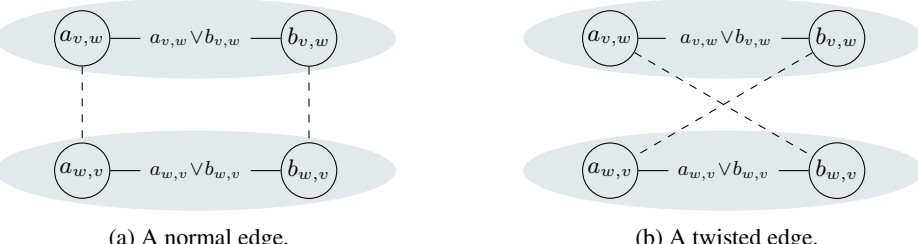

(a) A normal edge.   (b) A twisted edge.

Figure 4: Constructions in the formula $f_G$ and $\tilde{f}_G$ for an edge $\{v, w\} \in E(G)$. The edges are represented vertically, with the top ellipse corresponding to $v$'s side and the bottom to $w$'s side of the edge. Solid lines connect clauses to their literals. Dashed lines connect literals and their negations.

**Lemma B.7.** *Let $k$ be an odd integer and assume that $a_i \neq b_i$ for $1 \leq i \leq k$. $X_k$ is satisfied if and only if an even number of $a_i$'s (or equivalently $b_i$'s) are set to true.*

*Proof.* Let $T \subset \{1, \ldots, k\}$ be the set of indices $i$ such that $a_i$ is set to true. We start by proving that $X_k$ is satisfied whenever $|T|$ is even. The clauses in $D_k$ are satisfied whenever $a_i \neq b_i$, which is guaranteed by our assumption. Consider any clause $c_S = \left( \vee_{i \in S} a_i \right) \vee \left( \vee_{i \notin S} b_i \right)$. Since $k$ is odd, $|S|$ is even, and $|T|$ is even, $S \cup T$ is not a partition of $\{1, \ldots, k\}$. Hence, there is some index $i$ such that $i$ is in both sets or neither of the sets. If $i \in S \cap T$, then $a_i$ satisfies $c_S$. Else $i \notin S \cup T$, and $b_i$ satisfies $c_S$.

Conversely, suppose that $|T|$ is odd. Consider the clause $c_S$ with $S = \{1, \ldots, k\} \setminus T$. It is a clause because $|S|$ is even. It is unsatisfied, since $a_i = F$ for each $i \in S$ and $b_i = F$ for each $i \notin S$. $\square$

The following observation is a direct consequence of Lemma B.7 and Observation B.5:

*Observation* B.8. Assume the degree of every node in $G$ is odd. In any solution of $f_G$ or $\tilde{f}_G$, the number of $A$-literals set to true is even.

We say that a simple graph $G$ has an *even orientation* if there is an orientation of the edges such that all nodes have even outdegree. The following is a simple fact characterizing the existence of even orientations:

**Lemma B.9.** *Let $H$ be a simple connected graph. $H$ has an even orientation if and only if $m = |E(H)|$ is even.*

*Proof.* Let $d^{\text{out}}(v)$ denote the outdegree of a node $v$. It always holds that $\sum_{v \in V} d^{\text{out}}(v) = m$.

Assume there exists an even orientation of $H$. Then $d^{\text{out}}(v)$ is even for all $v$. Since the sum of even terms is always even, $m$ must be even.

Now assume that $m$ is even. Start with an arbitrary orientation of the edges. For any two vertices $v, w$ with odd outdegree, take an arbitrary path connecting $v$ and $w$ and reorient the edges on the path. This changes the outdegrees of $v$ and $w$ from odd to even, while not changing the parity of other nodes. Repeat this process until all nodes have even outdegree. Suppose that this is not possible, that is, we are left with a single node $v$ with odd outdegree. We have $d^{\text{out}}(v) = m - \sum_{w \in V \setminus \{v\}} d^{\text{out}}(w)$. The left side is odd, while the right side is even because $m$ and the terms of the sum are even. Hence, we can always find an even orientation. $\square$

Using this, we can prove the main lemma characterizing the satisfiability of $f_G$ and $\tilde{f}_G$:

**Lemma B.10.** *Let $G$ be a connected graph where all degrees are odd.[10] If $m = |E(G)|$ is even (odd), $f_G$ is satisfiable (unsatisfiable), and $\tilde{f}_G$ is unsatisfiable (satisfiable).*

---

[10]The proof can be generalized to a mix of odd and even degrees, but we chose to do this for simplicity of the argument.

*Proof.* Suppose that $m$ is even. $G$ has an even orientation by Lemma B.9. We can use this to extract a satisfying assignment of $f_G$ by setting $a_{v,w} = T$ for each outedge of $v$. This assignment satisfies every subformula by Lemma B.7. On the other hand, the twisted formula must have $m-1$ or $m+1$ $A$-literals true (Observation B.6), making it unsatisfiable (Observation B.8).

If $m$ is odd, $f_G$ is unsatisfiable by Observation B.8. On the other hand, the twisted edge allows us to set both $a$-literals of the edge to false, effectively removing the edge. The graph $G$ with $\{v, w\}$ removed has an even number of edges, so we can find a satisfying assignment by computing an even orientation of the remaining edges. □

*Proof of Theorem 5.3.* Let $G$ be a graph with odd degrees, where the size of the smallest separator is linear in $n$. We can use the construction of Ajtai (1994) for 3-regular expanders. The LCNs of $f$ and $f'$ are isomorphic to $X(G)$ and $\tilde{X}(G)$, respectively. These graphs are indistinguishable by Theorem B.3. By the above Lemma B.10, exactly one of $f_G$ and $\tilde{f}_G$ is satisfiable. Note that both $f_G$ and $\tilde{f}_G$ have at most 3 literals per clause. The number of variables is $n \cdot \Delta(G) = O(n)$ and the number of clauses $n \cdot (2^{\Delta(G)-1} + 1) = O(n)$. □

**Tseitin Formulas.** Interestingly, this construction is related to Tseitin formulas, which are known as hard instances for resolution refutation proofs (Urquhart, 1987).

**Definition B.11.** A Tseitin formula is constructed by taking a graph $G$ and a *charge function* $c$ : $V(G) \to \{0, 1\}$ labeling the vertices. Each edge $e \in E(G)$ is associated with a variable $x_e$. For each vertex, there is a constraint $\xi_v = \sum_{w \in N(v)} x_{\{v,w\}} = c(v) \mod 2$, meaning that the parity of the sum of variables of $v$'s edges is equal to the charge. The full formula is defined as $\wedge_{v \in V} \xi_v$.

It is known that a Tseitin formula is satisfiable if and only if the sum of charges is even. Hence, satisfiability is a global property of the graph. When the underlying graph is an expander (with small degrees), Tseitin formulas are known to be hard instances for resolution (Urquhart, 1987).

### B.3 Implications for WL-powerful architectures (proof of Lemma 5.4)

To prove Lemma 5.4, we first prove a more general statement (Lemma B.12). This uses the equivalence between $k$-WL and the pebbling games of Cai et al. (1992). Using the definition from (Kiefer, 2020), the *bijective $k$-pebble game* $\mathrm{BP}_k(G, H)$ is defined as follows. The game is played on two graphs $G$ and $H$. There are two players, a spoiler and a duplicator. The game proceeds in rounds. Each round is associated with a configuration $(\overline{u}, \overline{v})$ of pebbles on the two graphs, where $\overline{u} \in (V(G))^\ell$ and $\overline{v} \in (V(H))^\ell$, where $\ell \in [0, k]$. $G[\overline{u}]$ denotes the subgraph induced by the nodes in $\overline{u}$, where nodes carry the label of its position in the tuple $\overline{u}$. The initial configuration is the pair of empty tuples. In each round, the following actions are performed in order:

- The spoiler chooses an index $i \in [k]$
- The duplicator chooses a bijection $f : V(G) \to V(H)$
- The spoiler chooses $v \in V(G)$ and sets $w = f(v)$.

If $i \in [\ell]$, a pebble is moved from $u_i$ to $v$ in $G$ and from $v_i$ to $w$ in $H$. The new configuration is

$$((v_1, \ldots, v_{i-1}, v, v_{i+1}, \ldots, v_\ell), (w_1, \ldots, w_{i-1}, w, w_{i+1}, \ldots, w_\ell))$$

Otherwise, a new pebble is placed on $v$ in $G$ and on $w$ in $H$, and the new configuration is

$$((v_1, \ldots, v_\ell, v), (w_1, \ldots, w_\ell, w))$$

The spoiler wins if the induced, ordered subgraphs $G[\overline{u}]$ and $H[\overline{v}]$ are non-isomorphic. The duplicator wins if the spoiler never wins.

Let $G^{\overline{u}}$ denote the whole graph $G$, where nodes in $\overline{u}$ carry the label of their position in the tuple $\overline{u}$.

**Lemma B.12.** *Let $G$ and $H$ be two graphs indistinguishable by $k$-WL for some $k \geq 3$. Then, for every pebbling $\overline{u} \in (V(G))^t$, $t \leq k - 2$, there exists a pebbling $\overline{v} \in (V(H))^t$ such that the pebbled graphs $G^{\overline{u}}$ and $H^{\overline{v}}$ are indistinguishable by WL.*

*Proof.* By the equivalence between $k$-WL and the bijective $k$-pebble game (Cai et al., 1992)[11], the duplicator has a winning strategy in $\text{BP}_k(G, H)$. Starting from the empty configuration of $\text{BP}_k(G, H)$, let spoiler spend the first $t$ rounds to place $t$ pebbles on $G$ according to $\overline{u}$. Because the duplicator follows a global winning strategy in $\text{BP}_k(G, H)$, there exists a pebbling $\overline{v}$ on $H$ such that the configuration remains a partial isomorphism.

We now keep the $t$ pebbles fixed and consider the 2-pebble game played on the graphs $G^{\overline{u}}$ and $H^{\overline{v}}$, where the vertices of $\overline{u}, \overline{v}$ carry unique labels. Consider $\text{BP}_2(G^{\overline{u}}, H^{\overline{v}})$ which is equivalent to WL on these graphs. To argue that WL does not distinguish these graphs, we show that the duplicator has a winning strategy in $\text{BP}_2(G^{\overline{u}}, H^{\overline{v}})$.

We can simulate any spoiler play in $\text{BP}_2(G^{\overline{u}}, H^{\overline{v}})$ in $\text{BP}_k(G, H)$, keeping the first $t$ pebbles in place while using the $k - t \geq 2$ free pebbles to simulate the two pebbles in $\text{BP}_2$. Note that in $G^{\overline{u}}$, the vertices of $\overline{u}$ already carry unique labels, so the spoiler gains no additional power from placing a pebble on already labeled nodes, so we can assume spoiler only plays pebbles on vertices outside of $\overline{u}$. Each $\text{BP}_2$ round (spoiler picks index, duplicator picks bijection, spoiler picks a vertex in $G$) is then a legal $\text{BP}_k$ round. The duplicator responds with their winning strategy for $\text{BP}_k$. The spoiler never reaches a winning configuration in $\text{BP}_k(G, H)$, and hence cannot win in $\text{BP}_2(G^{\overline{u}}, H^{\overline{v}})$. Therefore the duplicator wins $\text{BP}_2(G^{\overline{u}}, H^{\overline{v}})$, which by the known equivalence is the same as WL failing to distinguish $G^{\overline{u}}$ and $H^{\overline{v}}$. □

**Lemma B.13.** *Let $f$, $\tilde{f}$ be formulas with LCNs indistinguishable by $k$-WL for some $k \geq 4$. For any partial assignment $\sigma$ of variables of $f$ with at most $\lfloor k/2 \rfloor - 1$ variables set, there is a corresponding partial assignment $\tilde{\sigma}$ of the variables of $\tilde{f}$ such that $\text{LCN}(\sigma(f))$ and $\text{LCN}(\tilde{\sigma}(\tilde{f}))$ are WL-indistinguishable.*

*Proof of Lemma 5.4.* The proof follows from Lemma B.12, which states that if two graphs are indistinguishable by $k$-WL, then any labeling of at most $k - 2$ nodes in one graph can be matched by a labeling of nodes in the other graph so that the resulting labeled graphs are WL-indistinguishable. We can think of assigned literals as being labeled with $\top$ or $\bot$. There are at most $2(\lfloor k/2 \rfloor - 1) \leq k - 2$ labeled literal nodes, since each variable corresponds to two literal nodes. By Lemma B.12, there exists a labeling of at most $k - 2$ nodes in $\text{LCN}(\tilde{f})$ such that the labeled graphs $\text{LCN}(\sigma(f))$ and $\text{LCN}(\tilde{\sigma}(\tilde{f}))$ are indistinguishable. The labels on these nodes determine a corresponding partial assignment ($\tilde{\sigma}$ cannot label two adjacent literals $\ell, \neg\ell$ with the same label (e.g. both $\top$), because such a labeling would be detectable by 1-WL as only existing in $\text{LCN}(\tilde{\sigma}(\tilde{f}))$) □

# C  GRAPH ISOMORPHISM COMPLETENESS OF DISTINGUISHING LITERAL-CLAUSE GRAPHS

To complement the indistinguishability results, we show that, in general, distinguishing LCNs is as hard as graph isomorphism. The graph isomorphism problem (GI) asks whether there is an edge-preserving bijection of the nodes. Formally, the decision problem is

$$\text{GI} = \{(G, H) : G, H \text{ are isomorphic graphs}\}$$

Let $\mathcal{G} = \{\text{LCN}(f) : f \in \text{3-SAT}\}$ be the set of LCNs of all 3-SAT formulas. We define the graph isomorphism problem on LCNs as the language

$$\text{GI}_{\text{SAT}} = \{(G, H) : G, H \in \mathcal{G} \text{ are isomorphic graphs}\}$$

We use the following formula to encode all relevant information about a graph in a CNF formula:

**Definition C.1** ($f_G$)**.** For each $v \in V(G)$, let $x_v$ be a variable. The edges of $G$ (and $|V(G)|$) can be encoded as a CNF formula $f_G = \left( \bigwedge_{\{v,w\} \in E(G)} (x_v \vee x_w) \right) \wedge \left( \bigwedge_{v \in V} x_v \right)$.

This formula is trivially satisfiable and not meaningful from a logical perspective, but it uniquely encodes $G$ up to isomorphism.

---

[11]Note that for $k \geq 3$, our definition of $k$-WL (which is more common in the machine learning literature) actually corresponds to $k - 1$-WL in (Cai et al., 1992; Kiefer, 2020).

*Observation* C.2. The LCN of $f_G$ is equal to $G$ with the following modifications. Each edge is subdivided with the corresponding clause node added in the middle. The original nodes correspond to the positive literals. Two leaf nodes corresponding to the negative literal and the unit clause is connected to each positive literal.

**Theorem C.3.** *The graph isomorphism problem on* LCN*s of* 3-SAT *formulas is equally hard as graph isomorphism on general graphs.*

*Proof.* GI $\leq$ GI$_{\text{SAT}}$: Let $G, H$ be two graphs and let $f_G, f_H$ be the corresponding formulas encoding the graph structure, as in Definition C.1. These two formulas are isomorphic iff $G, H$ are isomorphic: the two-variable clauses encode edges and the single-variable clauses encode nodes. The LCNs of $f_G, f_H$ are isomorphic iff the two formulas are isomorphic (Lemma 3.1).

GI$_{\text{SAT}}$ $<$ GI: Let $G_f, G_{f'}$ be two LCN*s*. This direction is easy, since $G_f$ and $G_{f'}$ are just two graphs. Note that graph isomorphism between edge-colored graphs can be reduced to graph isomorphism between uncolored graphs by replacing each edge with a special gadget that does not occur anywhere else in the graph, e.g. a $K_4$ since an LCN is tripartite. Specifically, for each literal-literal edge $\{x, \neg x\}$, remove the edge and add a $K_4$, connecting $x$ and $\neg x$ to the same vertex in the $K_4$. $\square$

## D  3-REGULAR SAT (PROOF OF THEOREM 5.1)

Recall that a bipartite graph is $(a, b)$-regular if all nodes in the left partition have degree $a$ and all nodes in the right partition have degree $b$.

*Observation* D.1. $(a, b)$-regular bipartite graphs with $n_A$ and $n_B$ nodes in the partitions are indistinguishable by WL.

We use the following lemma to manipulate the formula:

*Remark* D.2. $(x \vee y) \iff (x \vee y \vee z) \wedge (x \vee y \vee \neg z)$

**Lemma D.3** (Theorem 2.1 (Tovey, 1984) modified). *For any $\delta \geq 2$, given a 3-SAT formula $f$ with maximum literal degree $\Delta$, there is an equisatisfiable formula $f'$ with maximum literal degree $\delta$. The formula $f'$ has at most $n' = n + 4\Delta n$ variables and $m' = m + 4\Delta n$ clauses.*

*Proof.* Let $x$ be a variable in $f$ with $x$ or $\neg x$ appearing more than $\delta$ times. We break the variable into $s = d(x) + d(\neg x)$ variables $x_1, \ldots, x_s$, where $s \leq 2\Delta$. The constraint $x_1 = x_2 = \ldots, x_s$ is equivalent to $(x_1 \vee \neg x_2) \wedge (x_2 \vee \neg x_3) \wedge \cdots \wedge (x_s \vee \neg x_1)$. We use Remark D.2 on each clause to make them 3-regular. The end result uses $2s$ variables and has $2s$ clauses. Each original literal is used once in the constraint clauses, so it can be used $\delta - 1$ times outside of it. $\square$

**Lemma D.4.** *Given a 3-SAT formula $f$ where each literal appears in at most 3 clauses, there is an equisatisfiable 3-regular formula $f'$ where each literal is in exactly 3 clauses, with at most $n' = 5n$ variables and $m' = m + 9n$ clauses.*

*Proof.* Let $L_r \subseteq L$ be the literals in $f$ that appear in exactly 3 clauses. First, note that the number of edges in the literal-clause graph of $f$ is

$$3m = 3|L_r| + \sum_{\ell \in L \setminus L_r} d(\ell)$$

Here, the sum of degrees of non-regular literals must be a multiple of 3. We need to add $s = \sum_{\ell \in L \setminus L_r} 3 - d(\ell) = 3|L \setminus L_r| - \sum_{\ell \in L \setminus L_r} d(\ell)$ connections to make the literals 3-regular. As shown above, $s$ is a multiple of 3, so it is enough to show how to add 3 connections for literals $\ell_1, \ell_2, \ell_3$ (possibly some of these are equal). We introduce auxilliary variables $a, b, c, d$ and set $a = b = c = 1$, while the value of $d$ does not matter. We add the following clauses to form $f'$:

$$
\begin{array}{llll}
(\ell_1, a, \neg b), & (a, \neg c, d), & (b, \neg a, d), & (c, \neg b, d), \\
(\ell_2, \neg a, c), & (a, \neg c, \neg d), & (b, \neg a, \neg d), & (c, \neg b, \neg d), \\
(\ell_3, b, \neg c) & & &
\end{array}
$$

Note that all 9 clauses are satisfied by either $a, b$ or $c$. Also, all clauses are unique (even when $\ell_1 = \ell_2 = \ell_3$) and non-trivial (no clauses of type $x \vee \neg x$). All auxilliary literals appear exactly three

times. The set of solutions of $f'$ projected to the variables of $f$ is the same as the set of solutions for $f$.

The number of missing connections $s$ is at most $3n$, so the number of added variables and clauses is at most $4n$ and $9n$. □

# E  DISTINGUISHING RANDOM SAT INSTANCES (PROOF OF LEMMA 6.3)

This section uses the seminal results of Babai et al. (1980) on random graph isomorphism. To state our results, we need some related definitions. Let $\mathcal{K}$ be a class of graphs with $n$ vertices. A *canonical labeling algorithm* of $\mathcal{K}$ is an algorithm which assigns numbers $1, \ldots, n$ to each graph in $\mathcal{K}$, such that two graphs are isomorphic if and only if the labeled graphs coincide. Note that the labeling must be permutation invariant. Given a canonical labeling, we can test if two graphs are isomorphic in linear time by comparing the edges of the two graphs.

Babai et al. (1980) gave a canonical labeling algorithm for isomorphism testing on random graphs:

**Theorem E.1** ((Babai et al., 1980)). *There is a class of $n$-node graphs $\mathcal{K}$ and a linear-time algorithm that decides whether a given graph $G$ belongs to $\mathcal{K}$ and computes a canonical labeling of $\mathcal{K}$. The probability that a uniformly random $n$-node graph belongs to $\mathcal{K}$ is at least $1 - n^{-1/7}$, for large enough $n$.*

*Proof sketch.* In short, their algorithm computes a unique identifier for each node based on adjacency to high-degree nodes. For this to work, the top $r := 3 \log n / \log 2$ degrees must be unique. Assuming unique degrees for nodes $v_1, \ldots, v_r$ ordered by degree, the adjacency patterns to these nodes gives an $O(\log n)$-bit identifier $x_v$ to each node $v$, where $(x_v)_i = \mathbb{1}(v_i \in N(v))$. If the top degrees and the generated IDs are unique, this labeling is returned. Otherwise, the graph does not belong to $\mathcal{K}$. It can be shown that both conditions hold with probability at least $1 - n^{-1/7}$ over all $n$-node graphs.

It is known that the WL test identifies all graphs in $\mathcal{K}$. For clarity, we add the following lemma to make this formal:

**Lemma E.2.** *The WL test distinguishes any two graphs $G, H$ where $G \in \mathcal{K}$ and $H$ is any graph non-isomorphic to $G$.*

*Proof.* The first round labels $\chi_v^1$ of WL partition nodes by degree. The partition given by labels $\chi_v^2$ after the second round is clearly at least as fine as the partition given by $x_v$ in the algorithm of Theorem E.1. Hence, if $x_v$ is unique for each node, then so is $\chi_v^2$. Hence, the multiset of second-round labels is different for any $G \in \mathcal{K}$ and $H \notin \mathcal{K}$. In the third round, the unique labels encode all information about edges, so any differences in the adjacency between graphs in $\mathcal{K}$ is detected. □

## E.1  DISTINGUISHING INSTANCES EXTRACTED FROM RANDOM LIGS

Recall the literal-incidence graph representation of a CNF formula, where each literal is a node and two nodes are connected if they appear in the same clause. A principled way of extracting a CNF formula from a random literal-incidence graph is given by Wu & Ramanujan (2021) (see Lemma 6.2). We show that a CNF formula constructed this way is likely identified by the WL test.

**Theorem 6.3.** *A CNF formula extracted from a uniformly random literal-incidence graph with $n$ literals is identified by the WL test with probability at least $1 - (n)^{-1/7}$, over the choice of a LIG, for a large-enough $n$.*

*Proof.* Let $G$ be the corresponding literal-incidence graph. We show that the LCN is identified if $G$ is identified. Color refinement identifies $G$, i.e. each node gets a unique identifier (Theorem E.1), with probability at least $1 - (n)^{-1/7}$.

Consider color refinement on the LCN. We can ignore literal-literal edges – due to their different color, they only add expressivity. In two iterations, the information traveling from literals to literals via clauses on the LCN is exactly the same as in one iteration on $G$. By construction, the set of two-hop literal neighbors on the LCN (ignoring literal-literal edges) is exactly the same as the set of

neighbors in $G$. Hence, color refinement on the LCN produces unique identifiers for the literals (in at most twice the number of iterations). Since there are no duplicate clauses, clause nodes become identified by the unique set of literals they are connected to. Since all nodes have a unique identifier, the LCN is identified (as in Lemma E.2). □

## F EXPRESSIVITY VS. COMPUTATIONAL HARDNESS

A common misconception is that if a problem is computationally hard (e.g. NP-complete), then the WL hierarchy will also struggle to distinguish some instances of this problem. This is **not** the case. Distinguishability in GNNs, or equivalently the WL hierarchy (Xu et al., 2019; Morris et al., 2019) is a structural notion, whereas NP-hardness concerns the computational complexity of *deciding* a property. These notions are unrelated: there exists NP-hard problem families that the WL hierarchy fully **identifies**, i.e. distinguishes perfectly. For example, 4-WL distinguishes all instances of PlanarSAT (Theorem 6.1) even though PlanarSAT is NP-complete.

Conversely, our construction (Theorem 5.3) shows that there exists satisfiable and unsatisfiable formulas that remain indistinguishable to the full WL hierarchy. This indistinguishability arises because the satisfiability of these formulas does not localize in a way detectable by the WL refinement, not because SAT is computationally hard.

## G EXPERIMENTAL RESULTS

### G.1 DETAILS ON INSTANCE GENERATION

The default size of instances in G4SATBench is limited (200-400 variables for hard instances). To generate very large instances with number of variables in the thousands, we adapt the generation script for 3-SAT by slightly reducing the clause-to-variable ratio from the known satisfiability threshold (Crawford & Auton, 1996) of $\lfloor 4.258 \cdot n_V + 58.26 \cdot n_V^{-2/3} \rfloor$, where $n_V$ is the number of variables. This is known as the threshold number of clauses for 3-SAT instances, where the ratio of satisfiable to unsatisfiable formulas is approximately 50/50, and also where the hardest instances are typically found. To produce very large instances for the 3-SAT family, we change the multiplier from 4.258 to 4.158, which reduces the number of clauses slightly. This enables faster generation of satisfiable instances, while still maintaining approximately the same complexity.

### G.2 FULL RESULTS

Below are Tables 3 and 4, presenting the full results of our experiments on the SAT competition and G4SAT benchmark instances.

The competition instances were selected from the International SAT competition from years 2020 to 2025. All instances were initially satisfiable. Due to computational constraints, we only evaluated instances with size at most 10MB. Out of 448 evaluated instances, only 234 could be solved with the expressive power of WL. Across 69 instance families, 38 contained instances where WL is not expressive enough. Additionally, there were 72 instances where the evaluation timed out, because a solution was not found within 1 hour.

Table 3: Results on instances from the International SAT competition from years 2020 to 2025. All instances were initially satisfiable. $r_{\text{crit}}$ denotes the iteration in the Weisfeiler-Leman algorithm where the WL-partition-constrained formula becomes satisfiable (unsat if such an iteration does not exist). $r_{\text{converged}}$ is the number of iterations for the WL algorithm to converge. All values are reported as mean ± standard deviation.

| family | $r_{\text{crit}}$ | $r_{\text{converged}}$ | Variables | Clauses | Count |
|---|---|---|---|---|---|
| algebra | 43.00 ± 0.00 | 44.00 ± 0.00 | 12168 ± 0 | 55927 ± 0 | 1 |
| | unsat | 59.00 ± 0.00 | 45763 ± 21854 | 212025 ± 101562 | 3 |
| antibandwidth | unsat | 43.00 ± 0.00 | 56745 ± 0 | 177682 ± 0 | 1 |
| argumentation | 2.94 ± 0.44 | 4.31 ± 0.87 | 1266 ± 625 | 24764 ± 18863 | 16 |
| at-least-two-sol | unsat | 7.00 ± 0.00 | 2352 ± 0 | 219297 ± 0 | 2 |
| auto-correlation | 29.00 ± 0.00 | 32.00 ± 0.00 | 37869 ± 0 | 77287 ± 0 | 2 |
| battleship | unsat | 1.00 ± 0.00 | 762 ± 501 | 8466 ± 7770 | 5 |
| bitvector | unsat | 30.00 ± 0.00 | 11403 ± 0 | 33386 ± 0 | 1 |
| brent-equations | 31.11 ± 3.86 | 38.78 ± 12.34 | 72398 ± 9398 | 307082 ± 41212 | 9 |
| | unsat | 59.00 ± 0.00 | 78934 ± 5098 | 350649 ± 22681 | 2 |
| cardinality-constraints | 14.00 ± 0.00 | 18.00 ± 0.00 | 647 ± 0 | 3239 ± 0 | 1 |
| cellular-automata | unsat | 59.00 ± 0.00 | 42271 ± 0 | 248471 ± 0 | 1 |
| circuit-multiplier | unsat | 7.18 ± 0.40 | 1075 ± 50 | 20414 ± 1517 | 11 |
| coloring | unsat | 6.71 ± 4.61 | 15563 ± 28396 | 303330 ± 115583 | 14 |
| cover | unsat | 56.00 ± 6.00 | 162628 ± 199188 | 84331 ± 98439 | 4 |
| crafted-cec | 18.00 ± 1.10 | 19.00 ± 1.10 | 25109 ± 5971 | 75253 ± 17929 | 6 |
| cryptography | 15.74 ± 14.67 | 17.63 ± 14.34 | 41510 ± 29705 | 184548 ± 86532 | 19 |
| | unsat | 18.74 ± 13.38 | 19257 ± 37634 | 73907 ± 118053 | 23 |
| cryptography-ascon | unsat | 54.00 ± 0.00 | 158751 ± 58 | 373490 ± 258 | 7 |
| cryptography-simon | 18.00 ± 2.76 | 18.67 ± 2.80 | 3328 ± 599 | 11072 ± 2035 | 6 |
| discrete-logarithm | 17.50 ± 1.29 | 23.50 ± 4.43 | 14876 ± 3199 | 79248 ± 18265 | 4 |
| edge-matching | unsat | 13.67 ± 2.31 | 6615 ± 3633 | 73913 ± 44663 | 3 |
| edit-distance | 31.00 ± 0.00 | 32.00 ± 0.00 | 11083 ± 0 | 155964 ± 0 | 1 |
| ensemble-computation | 6.00 ± 0.00 | 8.00 ± 0.00 | 11607 ± 0 | 73804 ± 0 | 1 |
| fermat | 8.33 ± 0.58 | 11.00 ± 0.00 | 8203 ± 891 | 45927 ± 5155 | 3 |
| fixed-shape-random | 5.00 ± 0.00 | 5.00 ± 0.00 | 3486 ± 0 | 8496 ± 0 | 3 |
| fpga-routing | unsat | 9.50 ± 3.54 | 3331 ± 1783 | 42108 ± 27772 | 2 |
| generic-csp | unsat | 7.00 ± 0.00 | 840 ± 402 | 9858 ± 4852 | 2 |
| hamiltonian | 4.17 ± 0.51 | 5.44 ± 0.51 | 550 ± 45 | 4951 ± 408 | 18 |
| hamiltonian-cycle | unsat | 59.00 ± 0.00 | 33213 ± 5180 | 266845 ± 38797 | 9 |
| hw-model-check | unsat | 56.50 ± 3.54 | 119179 ± 39350 | 419860 ± 67060 | 2 |
| heule-folkman | unsat | 4.91 ± 0.30 | 16614 ± 1103 | 20147 ± 1492 | 11 |
| heule-nol | unsat | 8.60 ± 1.26 | 1419 ± 0 | 7832 ± 6 | 10 |
| hgen | 5.00 ± 0.00 | 5.00 ± 0.00 | 321 ± 1 | 1123 ± 3 | 14 |
| influence-max | 19.00 ± 2.83 | 21.50 ± 3.54 | 11323 ± 6549 | 62832 ± 36781 | 2 |
| knights-problem | unsat | 24.33 ± 1.15 | 107589 ± 25778 | 389927 ± 91758 | 3 |
| matrix-multiplication | 9.00 ± 0.00 | 10.00 ± 0.00 | 2396 ± 0 | 83733 ± 0 | 1 |
| max-const-part | 23.00 ± 0.00 | 52.00 ± 0.00 | 30252 ± 0 | 123402 ± 0 | 2 |
| maxsat-optimum | 26.64 ± 3.64 | 29.27 ± 2.49 | 22157 ± 5623 | 250704 ± 108568 | 11 |
| | unsat | 59.00 ± 0.00 | 27597 ± 8713 | 318348 ± 145483 | 7 |
| minimal-disagreement-parity | unsat | 59.00 ± 0.00 | 3176 ± 0 | 10283 ± 42 | 2 |
| minimal-superpermutation | 36.67 ± 4.04 | 40.00 ± 3.46 | 7885 ± 2006 | 26129 ± 5579 | 3 |
| | unsat | 59.00 ± 0.00 | 9075 ± 0 | 26255 ± 15479 | 2 |
| min-dis-parity | unsat | 8.73 ± 0.96 | 1042 ± 211 | 5861 ± 1234 | 15 |
| modcircuits | unsat | 7.00 ± 0.00 | 479 ± 0 | 123509 ± 0 | 1 |
| multiplier-circuits | 33.20 ± 12.49 | 37.80 ± 11.90 | 10955 ± 5629 | 43559 ± 22450 | 10 |
| planning | 14.00 ± 0.00 | 26.00 ± 0.00 | 439 ± 0 | 5423 ± 0 | 1 |
| | unsat | 25.00 ± 29.44 | 8666 ± 5382 | 74156 ± 36566 | 3 |
| poly-mult | unsat | 55.57 ± 9.07 | 57580 ± 31837 | 228133 ± 126348 | 7 |

| | | | | | |
|---|---|---|---|---|---|
| prime-factoring | 13.33 ± 10.69 | 16.17 ± 15.14 | 2305 ± 2557 | 16872 ± 5327 | 6 |
| prime-testing | 33.25 ± 14.15 | 34.00 ± 14.54 | 30102 ± 47706 | 98632 ± 134347 | 4 |
| purdom-instances | 22.33 ± 2.31 | 28.33 ± 11.85 | 4661 ± 327 | 18469 ± 1302 | 3 |
| pythagorean-triples | 7.00 ± 0.00 | 7.00 ± 0.00 | 3690 ± 4 | 13674 ± 75 | 2 |
| quasigroup-compl | 4.00 ± 0.00 | 4.00 ± 0.00 | 20940 ± 5970 | 436860 ± 149323 | 2 |
| | unsat | 5.00 ± 0.00 | 1890 ± 0 | 12666 ± 0 | 1 |
| random-circuits | unsat | 5.33 ± 1.15 | 3115 ± 231 | 184064 ± 13856 | 3 |
| random-csp | 3.60 ± 0.89 | 3.80 ± 0.45 | 799 ± 369 | 35723 ± 18087 | 5 |
| random-modularity | 3.00 ± 0.00 | 4.00 ± 0.00 | 2200 ± 0 | 9086 ± 0 | 2 |
| rbsat | 4.00 ± 0.00 | 4.00 ± 0.00 | 869 ± 206 | 55161 ± 22567 | 9 |
| scheduling | 12.14 ± 2.19 | 16.14 ± 3.48 | 16459 ± 3223 | 113224 ± 122825 | 7 |
| | unsat | 30.86 ± 13.37 | 35633 ± 29001 | 158119 ± 102896 | 28 |
| set-covering | 2.00 ± 0.00 | 4.00 ± 0.00 | 758 ± 51 | 31049 ± 3275 | 12 |
| sgen | unsat | 1.00 ± 0.00 | 220 ± 57 | 528 ± 136 | 2 |
| sliding-puzzle | 31.00 ± 0.00 | 33.00 ± 0.00 | 23029 ± 0 | 86273 ± 0 | 2 |
| social-golfer | 13.67 ± 1.15 | 16.67 ± 0.58 | 17469 ± 10785 | 103810 ± 72947 | 3 |
| | unsat | 7.33 ± 4.62 | 8098 ± 3956 | 91566 ± 65385 | 3 |
| software-verification | unsat | 54.00 ± 0.00 | 20965 ± 0 | 93999 ± 0 | 1 |
| ssp-0 | 12.00 ± 0.00 | 28.00 ± 0.00 | 50093 ± 0 | 214490 ± 0 | 1 |
| stedman-triples | 9.29 ± 1.54 | 10.14 ± 1.41 | 6225 ± 4578 | 142252 ± 118879 | 14 |
| subgraph-iso | 4.00 ± 0.00 | 5.00 ± 0.00 | 1065 ± 382 | 124130 ± 73773 | 3 |
| | unsat | 1.00 ± 0.00 | 598 ± 0 | 14076 ± 0 | 1 |
| sum-of-3-cubes | 11.50 ± 0.58 | 47.00 ± 13.88 | 35914 ± 6913 | 175274 ± 34488 | 4 |
| summle | unsat | 32.00 ± 0.00 | 102501 ± 12815 | 215796 ± 24955 | 17 |
| tensors | 14.92 ± 0.29 | 16.00 ± 0.00 | 3220 ± 0 | 12439 ± 0 | 12 |
| termination-analysis | 13.00 ± 0.00 | 17.00 ± 0.00 | 15092 ± 0 | 65248 ± 0 | 1 |
| tree-decomposition | unsat | 42.00 ± 0.00 | 74804 ± 0 | 393322 ± 0 | 2 |
| tseitin-formulas | 8.33 ± 1.15 | 8.67 ± 0.58 | 334 ± 21 | 2356 ± 147 | 3 |
| unknown | 9.33 ± 2.52 | 10.33 ± 3.21 | 2525 ± 1849 | 267927 ± 21151 | 3 |
| | unsat | 7.33 ± 1.15 | 7594 ± 7090 | 125672 ± 112383 | 3 |
| waerden | 2.00 ± 0.00 | 4.00 ± 0.00 | 242 ± 16 | 31376 ± 3731 | 2 |

Table 4: Results on the instances from the G4SAT benchmark. Instances are divided by family and difficulty, where difficulty follows the size-based categorization in (Li et al., 2024). The benchmark is designed so that all instances are challenging: hardness is controlled via the clause-to-variable ratio, which is fixed close to the satisfiability threshold for 3-SAT, and analogously for families encoding combinatorial problems. All instances were initially satisfiable. $r_{\mathrm{crit}}$ denotes the WL iteration where the WL-partition constrained formula becomes satisfiable (unsat if such an iteration does not exist). $r_{\mathrm{converged}}$ is the number of iterations for the WL algorithm to converge. On average, a few iterations is enough for WL to sufficiently distinguish literals, except for a few outlier instances in the $k$-clique, $k$-vertex cover and ca families. The PS family is omitted due to problems with the generation script. All values are reported as mean ± standard deviation.

| family | difficulty | $r_{\mathrm{crit}}$ | $r_{\mathrm{converged}}$ | Variables | Clauses | Count |
|---|---|---|---|---|---|---|
| 3-sat | easy | 2.97 ± 0.18 | 3.68 ± 0.47 | 26 ± 9 | 119 ± 36 | 1000 |
| | medium | 3.00 ± 0.04 | 3.92 ± 0.28 | 119 ± 47 | 509 ± 198 | 1000 |
| | hard | 3.00 ± 0.00 | 4.00 ± 0.00 | 250 ± 29 | 1065 ± 125 | 100 |
| | hard+ | 3.00 ± 0.00 | 4.00 ± 0.00 | 921 ± 48 | 3775 ± 196 | 24 |
| | hard++ | 3.08 ± 0.28 | 4.00 ± 0.00 | 5001 ± 62 | 20504 ± 256 | 25 |
| k-clique | easy | 4.12 ± 0.73 | 6.26 ± 0.83 | 33 ± 13 | 543 ± 426 | 960 |
| | | unsat | 6.00 ± 0.78 | 22 ± 7 | 217 ± 194 | 40 |
| | medium | 4.11 ± 0.52 | 6.33 ± 0.95 | 68 ± 17 | 2156 ± 960 | 999 |
| | | unsat | 6.00 ± 0.00 | 45 ± 0 | 939 ± 0 | 1 |
| | hard | 4.00 ± 0.00 | 6.00 ± 0.00 | 114 ± 20 | 5554 ± 1718 | 100 |
| k-domset | easy | 4.11 ± 0.71 | 5.61 ± 0.93 | 39 ± 12 | 329 ± 186 | 1000 |
| | medium | 4.33 ± 0.58 | 5.50 ± 0.72 | 88 ± 18 | 1647 ± 687 | 1000 |
| | hard | 4.42 ± 0.67 | 5.54 ± 0.69 | 137 ± 22 | 3986 ± 1315 | 100 |
| k-vercov | easy | 4.75 ± 1.14 | 6.18 ± 1.38 | 39 ± 13 | 358 ± 245 | 993 |
| | | unsat | 4.00 ± 0.00 | 26 ± 8 | 159 ± 108 | 7 |
| | medium | 4.94 ± 1.00 | 5.88 ± 1.08 | 96 ± 20 | 2052 ± 936 | 1000 |
| | hard | 5.00 ± 1.01 | 5.80 ± 0.85 | 179 ± 25 | 7198 ± 2159 | 100 |
| sr | easy | 2.00 ± 0.05 | 3.00 ± 0.06 | 25 ± 9 | 146 ± 54 | 1000 |
| | medium | 2.01 ± 0.12 | 3.00 ± 0.00 | 118 ± 47 | 644 ± 249 | 1000 |
| | hard | 2.05 ± 0.22 | 3.00 ± 0.00 | 299 ± 62 | 1613 ± 343 | 100 |

