# OpenReview forum: "On the Expressive Power of GNNs for Boolean Satisfiability"
_ICLR.cc/2026/Conference — ICLR 2026 Poster_

### Official Review · Reviewer_2HM5 · 2025-10-28

**Soundness:** 3
**Presentation:** 3
**Contribution:** 3
**Rating:** 6
**Confidence:** 3

**Summary:**

The paper analyzes the expressive capabilities of Graph Neural Networks (GNNs) in the context of SAT solving. The authors particularly look at the Literal-clause graphs with negation connections (LCNs), one of the standard representation used in GNNs for SAT solving. The authors then theoretically prove which families of problems GNNs can and cannot distinguish, using the Weisfeiler-Lehman (WL) test, a well known expressivity analysis technique for GNNs. With experiments, the authors assess whether WL-powerful architectures are, in principle, capable of predicting satisfying assignments on a wide range of benchmark instances including randomly generated and competition benchmark instances. The results show that while almost all random instances are WL-distinguishable, many SAT competition instances are indistinguishable, showing the limitations of GNNs for industrial SAT solving.

**Strengths:**

- The theoretical results imply that with LCNs, any Message Passing Neural Network based models cannot fully distinguish between SAT and UNSAT instances for certain families of problems such as 3-SAT, even when given partial assignment information.
- The authors conduct experimental results to determine whether instances in a benchmark set are WL-distinguishable or not, with the experiments being done on a wide range of benchmarks sets.
- The paper provides insights into the limitations of GNNs for SAT solving, which is an important topic given the recent interest in using machine learning for combinatorial optimization problems.

**Weaknesses:**

- The theoretical results are limited to LCNs, due to the limitations of expressivity analysis with WL tests as stated in the Appendix. While I understand LCNs to be a common approach of using GNNs for SAT solving such as in NeuroSAT, VCGs are also widely used such as in [1], and the paper would greatly benefit from discussing their insights, other than just briefly stating that it is not possible to perform expressivity analysis on them. I personally think that as VCGs incorporate polarity information directly into the graph edges, it allows them to distinguish graphs for formulas such as those in Figure 2.
- The random instances from G4SATBench may not be as informative as ones from the SAT competition. To my understanding, this comes down to whether the benchmark was able to randomly produce any problem that is difficult to solve. If performing evaluations on random instances were the objective, the random track from the 2018 SAT competition may be better suited in this regard.
- I see no reason to limit the experimental results to the 2024 SAT competition benchmarks. Including benchmarks from previous years would only strengthen the experimental contributions.

[1] Yolcu, E., & Póczos, B. (2019). Learning Local Search Heuristics for Boolean Satisfiability. In H. M. Wallach, H. Larochelle, A. Beygelzimer, F. d'Alché‑Buc, E. B. Fox, & R. Garnett (Eds.), Advances in Neural Information Processing Systems 32 (NeurIPS 2019) (pp. 7990–8001).


Minor issues:
- The graph visualization in Figure 1 does not match up with the formula given. I believe that C6 should be C1, if we assign clause numbers in order of appearance.
- Figure 2 seems to be in the middle of the references section.
- Citations are not formatted correctly; they are missing parentheses. For example "variables Heule et al. (2024)" at L123 should be "variables (Heule et al., 2024)".
- In L132 and L137, $\chi^\ell(v) := (\chi^{\ell-1}(v), \{ \{ \chi^{\ell-1}(v) : w \in N(v) \} \})$ should be written as $\chi^\ell(v) := (\chi^{\ell-1}(v), \{ \{ \chi^{\ell-1}(w) : w \in N(v) \} \})$ (the $v$ should be $w$ in the inner set).
- In L412, it says "the only family with some formulas that could not be solved by WL is the k-clique", but Table 4 clearly shows k-vercov as another family that WL could not fully solve.

**Questions:**

- Is it possible at all to extend the WL-test to allow for expressivity analysis of VCGs?
- Are there other suitable graph representations of SAT problems that would be better than LCNs in terms of expressivity? What would the best representation look like, if any?
- Why were harder instances (hard+, hard++) provided only for 3-SAT? Is it not possible to generate harder instances for other problems as well?
- The experimental analysis were done on SAT instances only. Would it be possible to extend the analysis to UNSAT instances?

---

> ### Author Response · Authors · 2025-11-13
> **Author response**
>
> We thank the reviewer for their thoughtful and constructive review.
>
> ## Formula Representation
>
> > The theoretical results are limited to LCNs... , VCGs are also widely used... I personally think that as VCGs incorporate polarity information directly into the graph edges, it allows them to distinguish graphs for formulas such as those in Figure 2.
>
> We chose the Literal-Clause graph with Negation connections (LCN) because it is the most expressive commonly used representation, whose graph structure corresponds one-to-one with a formula. In contrast, for the Variable-Clause Graphs (VCGs), there are actually $2^n$ different representations, corresponding to essentially the same formula. For example $(\neg x \vee y) \wedge x$ and $(x \vee y) \wedge \neg x$ have exactly the same set of solutions, after flipping the value of $x$. This *invariance under variable flips* is naturally respected in the LCN and makes it more suited for GNNs. This is analogous to requiring permutation equivariance/invariance. There are some more details in Appendix A.
>
> > VCGs are also widely used such as in [1], and the paper would greatly benefit from discussing their insights
>
> We agree and will add this in the updated version.
>
> ## Random Instances
>
> > The random instances from G4SATBench may not be as informative as ones from the SAT competition. To my understanding, this comes down to whether the benchmark was able to randomly produce any problem that is difficult to solve.
>
> > Why were harder instances (hard+, hard++) provided only for 3-SAT? Is it not possible to generate harder instances for other problems as well?
>
> All instances in the G4SATBench are generated to be hard in the classical sense. For 3-SAT, the benchmark selects instances around the *satisfiability treshold* [1], where formulas are known to be hard to solve. The labels *easy, medium, hard* are from G4SATBench, and only refer to the **size of the instance**. The benchmark construction keeps difficulty comparable across sizes.
>
> For our experiments, we extended the range of sizes for 3-SAT, and labeled the families hard+, hard++ (respecting the naming convention in G4SATBench). We inteded to generate analogous extensions for other random families, but generating and evaluating the instances took too long (several hours per instance) to do on a larger scale (see Appendix F.1).
>
> [1] James M. Crawford and Larry D. Auton. Experimental results on the crossover point in random 3-sat. Artificial Intelligence, [URL](https://www.sciencedirect.com/science/article/pii/0004370295000461). Frontiers in Problem Solving: Phase Transitions and Complexity.
>
> ## Questions
>
> > Is it possible at all to extend the WL-test to allow for expressivity analysis of VCGs?
>
> It is possible to extend our results to VCGs, but the main difficulty is interpretability. In the above example, flipping all occurrences of a variable can make two VCGs different, even if the original formulas were the same. Still, we expect analogous pairs of indistinguishabilite instances to exist for VCGs: for example, pairs of SAT/UNSAT instances with VCGs indistinguishable to n-WL should also exist.
>
> > Are there other suitable graph representations of SAT problems that would be better than LCNs in terms of expressivity? What would the best representation look like, if any?
>
> Appendix A contains an overview of commonly used graph representations for SAT. The LCN is the only common representation that is both lossless and stable. Other representations either discard information (literal-incidence, variable-incidence graphs) or break invariances (VCG).
>
> > I see no reason to limit the experimental results to the 2024 SAT competition benchmarks.
>
> We agree and are running the experiments for earlier years. We will include the results in the updated version.
>
> > The experimental analysis were done on SAT instances only. Would it be possible to extend the analysis to UNSAT instances?
>
> The current setup is for predicting a *satisfying assignment*, which only makes sense for SAT instances. One could instead test *graph-level distinguishability* of UNSAT and SAT instances. The problem for competition instances is that there are too few SAT/UNSAT pairs with identical size. For random instances, we observed that WL commonly assigns each variable a unique label. This implies that WL also distinguishes the instances on the graph level [2]. For these reasons, we focused on satisfiable formulas.
>
> [2] László Babai, Paul Erdös, and Stanley M. Selkow. Random graph isomorphism. SIAM Journal
> on Computing, 9(3):628–635, 1980. doi: 10.1137/0209047. [URL](https://doi.org/10.1137/0209047)
>
>
> **Minor issues:**
> Thank you for pointing these out. We will fix them in the updated version.
>
> We hope we have addressed your questions clearly, and we again thank the reviewer for their effort.

---

### Official Review · Reviewer_4PqT · 2025-10-29

**Soundness:** 3
**Presentation:** 3
**Contribution:** 3
**Rating:** 6
**Confidence:** 3

**Summary:**

The power of GNNs can be understood by studying WL-testing, which is basically a form of graph isomorphism testing. On a high level, GNNs cannot distinguish some graphs from each other and this is reliant on how the weisfeiler lehman kernel behaves on these graphs. The paper shows that even if GNNs were "maxed out" on the WL power hierarchy, there would be SAT instances that would be difficult for them since it is difficult for the WL process itself (this is actually not surprising, given that SAT is a hard problem) but more interestingly, it creates these hard instances constructively, which is better than showing that they exist. Empirically, it then studies real SAT instances. Probably as expected, random SAT instances can fall to WL , and contest SAT instances don't fall as easily.

**Strengths:**

The paper's contribution is mostly theoretical. One of the best aspects is that they actually construct the hard family of instances, which is likely to be useful for further work more than a simple proof of existence. Also, although the paper is about GNNs on paper, most of the results are about the WL kernel's suitability for SAT, which is technically a broader problem. In reality, the paper is about finding the "blind spots" of the WL kernel as a method.

**Weaknesses:**

The empirical study is fairly surface level and misses the mark. We already expect that random instances are easy, contest instances are harder, and hard-by-construction instances are computationally hard. But the main problem is as follows. The power of a GNN is only upper bounded by the power of a WL test. Showing how well the WL test performs on various SAT instances, then, is meaningful only in one direction - if it fails, certainly we don't expect GNNs to succeed, but if it succeeds or performs in an intermediate manner, we *cannot actually confirm* that GNNs will follow ! Thus, the construction of the hard instances is meaningful "back to GNNs" though WL is being studied, but the studies on empirical "actually existing" instances are statements that only mean something wrt the WL kernel, not GNNs themselves.

**Questions:**

It seems to me that the paper is actually about the suitability of the WL kernel to SAT, and can be thought of better that way. Do you agree ? From a GNN point of view, the empirical section is actually not meaningful (as pointed out under weaknesses)

---

> ### Author Response · Authors · 2025-11-13
> **Author response**
>
> We thank the reviewer for their comments and suggestions. We address the concerns below.
>
> > The paper shows that even if GNNs were "maxed out" on the WL power hierarchy, there would be SAT instances that would be difficult for them since it is difficult for the WL process itself (this is actually not surprising, given that SAT is a hard problem)
>
> It is important to clarify that **computational hardness and WL indistinguishability are unrelated notions**. There are NP-hard problem families where all instances can be distinguished. For example, 4-WL distinguishes all instances in PlanarSAT (Theorem 6.1), even though PlanarSAT is NP-complete. Thus, the existence of satisfiable/unsatisfiable pairs of formulas is not a consequence of SAT being hard. Rather, indistinguishability arises because satisfiability of these formulas does not localize in a way that the WL-hierarchy can capture.
>
> ## Empirical Study
>
> > We already expect that random instances are easy, contest instances are harder, ..
>
> This is actually not the case in general. Random SAT formulas can also be hard in the classical sense, and in fact, G4SATBench is designed to only generate **hard random instances**. As discussed in our response to Reviewer 2HM5, these instances are chosen near phase transitions which makes them computationally hard.
>
> > Showing how well the WL test performs on various SAT instances, then, is meaningful only in one direction - if it fails, certainly we don't expect GNNs to succeed, but if it succeeds or performs in an intermediate manner, we _cannot actually confirm_ that GNNs will follow !
>
> We fully agree that WL analysis only gives **necessary**, but not sufficient, conditions for GNN learning. This is explicitly noted in the paper (e.g. lines 378 and 478). The goal of the empirical section is not to predict GNN performance, but to validate the theoretical claim that different SAT families place very different demands on expressive power.
>
> The WL test is a standard tool in GNN expressivity analysis. The resulting statements describe what architectures can or cannot represent, not what they will learn in practice.
>
> ## Questions
>
> > It seems to me that the paper is actually about the suitability of the WL kernel to SAT, and can be thought of better that way. Do you agree ?
>
> We appreciate this question and would like to clarify terminology. The *Weisfeiler Leman test* (Definition 2.1) and *WL kernel* are conceptually related, but they are not the same thing. The WL test (and its higher order variants) upper bound the expressive power of GNNs. The *WL kernel* is a specific graph kernel based on the same refinement procedure.
>
> Interpreting our work as characterizing "blind spots" of the WL test is accurate, and in fact our goal. The reason we frame the paper in terms of GNNs is that GNN-based SAT solving is the practical setting where these expressivity limits matter. While WL kernels could theoretically be analyzed in the same way, we are not aware of any work applying graph kernels to SAT solving.
>
> We hope that these clarifications address the reviewer's concerns, and we thank the reviewer again for their time and engagement.

---

### Official Review · Reviewer_cb9V · 2025-10-31

**Soundness:** 4
**Presentation:** 4
**Contribution:** 3
**Rating:** 6
**Confidence:** 4

**Summary:**

The paper analyzes the expressive limits of GNNs on SAT problems using a literal–clause–negation (LCN) graph representation. It proves that even the full Weisfeiler–Lehman (WL) hierarchy cannot, in general, distinguish satisfiable from unsatisfiable 3-SAT formulas, via a parity-based construction. The authors further show that this limitation persists in sequential variable-assignment settings, linking theory to neural SAT solvers. They complement the negative results with positive findings (e.g., PlanarSAT identifiable by 4-WL) and empirical WL-color diagnostics on random and industrial benchmarks. The work provides a clear, rigorous theoretical contribution, though experiments remain illustrative rather than learning-based.

**Strengths:**

The presentation is clear and well-organized with logical progression from simple examples to full results, making complex theoretical concepts accessible while providing complete proofs in appendices. The analysis is nuanced, including both impossibility and positive results (e.g., PlanarSAT separability, random instances).
The authors are transparent about the scope of their results, clearly acknowledging that their experiments test only necessary conditions for expressivity and do not demonstrate actual learning performance.
The main theoretical contribution (Theorem 5.3) is rigorous, proving that even the full n-WL hierarchy cannot distinguish SAT/UNSAT formulas in general, and this limitation transfers to practical sequential solvers (Corollary 5.5), showing that even after Θ(n) variable assignments, the indistinguishability persists - meaningfully connecting fundamental expressivity limits to realistic GNN-based SAT solving approaches. The link to Tseitin formulas from proof complexity is noteworthy.

**Weaknesses:**

The paper lacks any trained GNN experiments, making it unclear whether the theoretical limitations observed actually translate into performance gaps in practice.  The experiments are purely diagnostic and do not involve any trained GNNs, leaving the practical impact of the theoretical limits untested.

Lacks an explanation for why industrial instances require higher expressivity than random ones. The paper does not discuss possible ways to overcome the identified expressivity limits, leaving the reader without guidance on how future GNN architectures might address these shortcomings.

**Questions:**

Given that the experiments are WL-diagnostic rather than learned, do you plan to test whether these expressivity limits manifest in actual trained GNN solvers?

Given that even n-WL cannot distinguish SAT/UNSAT in general, what architectural modifications beyond standard MPNNs (e.g., augmentations, auxiliary features, hybrid approaches) do you believe could address these expressivity bottlenecks while remaining scalable?

---

> ### Author Response · Authors · 2025-11-13
> **Author response**
>
> We thank the reviewer for their careful review and constructive feedback.
>
> ## Industrial vs. Random Instances
>
> > [The paper] lacks an explanation for why industrial instances require higher expressivity than random ones.
>
> Industrial formulas often encode problems from highly structured domains, like hardware verification or cryptography. These domains naturally produce regular or highly symmetric formulas. This creates large WL-equivalence classes, which exposes the lack of expressivity in WL-bounded GNN approaches.
>
> In contrast, random instances behave similarly to random graphs in classical isomorphism testing, where almost all nodes become distinguishable after very few WL iterations [1]. This does not necessarily mean the instances are easier to solve, but it does mean they don't stretch the expressivity limits of GNNs.
>
> [1] László Babai, Paul Erdös, and Stanley M. Selkow. Random graph isomorphism. SIAM Journal
> on Computing, 9(3):628–635, 1980. doi: 10.1137/0209047. [URL](https://doi.org/10.1137/0209047)
>
> ## Experiments Involving Learning
>
> > The paper lacks any trained GNN experiments, making it unclear whether the theoretical limitations observed actually translate into performance gaps in practice.
>
> > Given that the experiments are WL-diagnostic rather than learned, do you plan to test whether these expressivity limits manifest in actual trained GNN solvers?
>
> As shown in Section 6 and 7, random instances are easier than industrial instances from an expressivity standpoint. Ideally, we would like to train and test GNNs on industrial instances to assess how big the expressivity limits are in practice. However, current datasets make this very challenging. Industrial SAT instances are typically too large for training, and the number of available instances is far smaller than what is required.
>
> One of our motivations is therefore to highlight this gap: theoretical expressivity limitations are relevant, but the field currently lacks suitable non-random datasets that reflect this (Section 8).
>
> ## Expressive Architectures
>
> > The paper does not discuss possible ways to overcome the identified expressivity limits, leaving the reader without guidance on how future GNN architectures might address these shortcomings.
>
> > Given that even n-WL cannot distinguish SAT/UNSAT in general, what architectural modifications beyond standard MPNNs (e.g., augmentations, auxiliary features, hybrid approaches) do you believe could address these expressivity bottlenecks while remaining scalable?
>
> This is an important question. Many current approaches aim to predict satisfiability, or even a full satisfying assignment, in a single forward pass. This is arguably too difficult. Our view is that learning-based SAT could work better in the sequential setting, mirroring classical solvers.
>
> Within such a framework, simple techniques like randomized symmetry breaking (given a formula $f$, run the solver with $x=$True and $x=$False) may help to overcome some expressivity issues. The idea is to *get the solver started*, for example when the initial formula is a regular graph. At the same time, Theorem 5.3 and Lemma 5.4 show that even after a large number of variable assignments, the expressivity of WL may still be insufficient.
>
>
>
> We hope these answers address the reviewer's questions, and again thank the reviewer for their careful review and feedback.

---

### Official Review · Reviewer_ZqMt · 2025-11-06

**Soundness:** 3
**Presentation:** 3
**Contribution:** 1
**Rating:** 4
**Confidence:** 5

**Summary:**

This paper analyzes the expressiveness power of graph neural networks (GNNs) for the Boolean constraint satisfiability problems. The analysis is based on the observation that the power of GNNs is bounded by Weisfeiler-Leman (WL) test, which is designed to test graph isomorphism. Due to such fundamental limitation, GNNs are proved to be insufficient to solve NP-Complete problems. Furthermore, empirical evaluations are performed on the G4SAT benchmark and SAT Competition 2024.

**Strengths:**

- this paper consolidates many known theoretical results about GNNs and WL test as well as empirical datasets
- background knowledge about SAT solving, GNNs, graph isomorphism, and WL tests are carefully illustrated

**Weaknesses:**

- the proposed analysis is not novel, since all reported results are either well-known or immediately implied by previous works. Particularly, Xu et al. (2019) has pointed out that GNNs are bounded by the WL-test, which cannot even solve the graph isomorphism problems that are not as strong as the NP-Complete problems like Boolean satisfiability.
- the presented theoretical analysis is largely rephrasing previous known results therefore does not contribute new insights
- there are some empirical observations regarding the number iterations for WL to converge, however, which does not provide meaning guidance on solving satisfiability problem with GNNs

**Questions:**

Besides consolidating previous known results, which is a meaningful contribution in the perspective of literature survey, what new insights either theoretical or empirical does this paper provide?

---

> ### Author Response · Authors · 2025-11-13
> **Author response**
>
> We thank the reviewer for their comments and are happy to clarify several points below.
>
> > the proposed analysis is not novel, since all reported results are either well-known or immediately implied by previous works. Particularly, Xu et al. (2019) has pointed out that GNNs are bounded by the WL-test, which cannot even solve the graph isomorphism problems that are not as strong as the NP-Complete problems like Boolean satisfiability.
>
> This statement is based on a misunderstanding of what standard expressivity results imply. The notions of distinguishability and hardness are in general unrelated. In more detail, the fact that GNNs and the WL hierarchy cannot distinguish some graphs **does not imply** that they cannot distinguish some SAT/UNSAT instances. For example, 4-WL cannot solve graph isomorphism but can *fully distinguish* all instances of the NP-complete PlanarSAT (Theorem 6.1).
>
> > the presented theoretical analysis is largely rephrasing previous known results therefore does not contribute new insights
>
> > Besides consolidating previous known results, which is a meaningful contribution in the perspective of literature survey, what new insights either theoretical or empirical does this paper provide?
>
> As explained above, our main result (Theorem 5.3) is not related or implied by prior WL expressivity bounds. In addition, we provide
> - A limit theorem for sequential GNN solvers (Lemma 5.4, Corollary 5.5), connecting the theoretical result on k-WL indistinguishability to practical limitations
> - Positive results showing that random instances are often distinguished (Theorem 6.3), and—following known results on planar graphs—that all PlanarSAT instances are distinguished by 4-WL. The latter is also an interesting example showing that computational hardness and distinguishability are unrelated.
> - Empirical evidence confirming our theoretical results: random instances are easier from an expressivity point of view, while industrial instances often exceed WL capacity
>
> > there are some empirical observations regarding the number iterations for WL to converge, however, which does not provide meaning guidance on solving satisfiability problem with GNNs
>
> While not the main focus of the paper, our experiments confirm the theory that random instances are easier from the expressivity point of view. They may still be hard to solve in the usual sense, but distinguishing them does not require much expressive power. This suggests that random benchmarks do not capture the structural challenges in GNN-based SAT solving, and relying on them for benchmarking risks overlooking the real bottlenecks.
>
> We hope the above clarifications have helped. Since the main criticism appears to rely on an incorrect connection between distinguishability and computational hardness, we respectfully ask the reviewer to reconsider their evaluation.

---

> > ### Author Response · Authors · 2025-11-25
> >
> > We have revised the manuscript to include Appendix F, which clarifies the relation between WL-indistinguishability and computational hardness. If you have any further thoughts on the points clarified here, we would be glad to hear them.

---

### Author Response · Authors · 2025-11-19
**List of changes**

We are grateful to the reviewers for their time and valuable feedback.

We have revised the manuscript according to the suggestions from reviewers. The changes are summarized below:
- **Clarified the distinction between expressivity and computational hardness**: To address a common point of confusion, we added Section F in the appendix to explain how expressivity interacts with NP-complete problems like Boolean Satisfiability. The new section clarifies that expressivity and computational hardness are conceptually distinct, and that NP-hardness does not imply the existence of indistinguishable instances. We hope this clarification helps to contextualize our main result.
- **Improved experimental results**: We extended the experiments to include more years of the International SAT competition (2020–2025). The new evaluation covers both families already present in the original submission, as well as new problem families. The full results are shown in Table 3. These results provide additional evidence that industrial instances place higher demands on expressive power.
- **Clarified the choice of graph encoding**: We expanded Appendix A to make the motivation for using Literal Clause Graphs clearer. In particular, we explain why LCGs are preferable over Variable Clause Graphs from a graph-learning perspective.
- **Updated related work**: As suggested by Reviewer 2HM5, we've added a reference to Learning Local Search Heuristics for Boolean Satisfiability.
- **Format**: We corrected an issue with citation formatting.
- **Minor corrections**: Fixed typos and layout issues.

We hope these revisions have addressed all concerns. Thank you again for your thoughtful comments and interest in our work.

---

### Meta-Review · Area_Chair_YibA · 2025-12-19

**Summary:**

The paper presents a rigorous theoretical and empirical analysis of the expressive power of Graph Neural Networks (GNNs) for Boolean Satisfiability (SAT). Using the Weisfeiler-Leman (WL) framework, the authors prove a significant negative result: even the full WL hierarchy cannot generally distinguish between satisfiable and unsatisfiable 3-SAT instances. This limitation extends to sequential variable assignment settings, meaning that even after a large number of assignments, indistinguishability can persist for WL-bounded architectures.

The authors balance these negative results with positive findings for specific families, such as PlanarSAT (fully identified by 4-WL) and random instances (largely identifiable by WL). Empirically, they demonstrate that while random benchmarks are easily distinguished, industrial instances from the SAT competition often exceed the expressivity limits of standard GNN representations.

My own concerns as as follows:

- Empirical Gap: The most prominent concern is the lack of "learning-based" experiments. The authors use "WL-diagnostic" tests to check necessary conditions for expressivity, but they do not train GNNs to solve SAT, leaving practical performance impacts partially untested.
- Graph Representation Limitations: The focus is primarily on the Literal-Clause Graph with Negation (LCN). While well-justified, the exclusion of Variable-Clause Graphs (VCGs) from the primary theoretical analysis was a point of contention for some reviewers.

While I still lean towards an acceptance, this paper may be more suitable for other venues (e.g., SAT conference)

**Reviewer Concerns:**

Addressed by Rebuttal:

- Conceptual Clarity on Hardness: Reviewers ZqMt and 4PqT suggested the results were unsurprising because SAT is NP-hard. The authors successfully clarified that computational hardness and structural distinguishability are unrelated, noting that PlanarSAT is NP-complete but distinguishable by $4$-WL. They added Appendix F to formalize this.
- Dataset Scope: Reviewer 2HM5 questioned the reliance on the 2024 SAT competition. The authors addressed this by extending experiments to cover competitions from 2020–2025 (Table 3).
- LCN vs. VCG: The authors provided a robust technical defense for using LCN, arguing it is the only common representation that is both lossless and stable under variable flips, whereas VCG can produce non-isomorphic encodings for the same formula.

Outstanding Concerns:

- Lack of Trained GNNs: Reviewer cb9V remains concerned that the theoretical limits may not fully translate to performance gaps in practice without trained models. The authors' defense is practical—industrial instances are too large for current training methods—but this leaves the empirical section as a "diagnostic" validation rather than a performance benchmark.

**Reviewer Scores:**

- ZqMt (possibly from 4 to 6): The author's rebuttal corrected a significant misunderstanding regarding the relationship between NP-hardness and WL-distinguishability.
-  cb9V (6 to 6): Though the reviewer found the theory "excellent," the lack of trained GNNs remains a persistent weakness in their view
- 4PqT (6 to 6): The reviewer acknowledged the constructive proof is "useful for further work" despite the surface-level empirical study.
- 2HM5 (6 to 6): The authors directly addressed this reviewer's request for more experiments and a discussion on VCGs.

---

### Decision · Program_Chairs · 2026-01-26

Accept (Poster)